# FSL-SAGE: Accelerating Federated Split Learning via Smashed Activation Gradient Estimation

Srijith Nair [1]   Michael Lin [1]   Peizhong Ju [2]   Amirreza Talebi [3]   Elizabeth Serena Bentley [4]   Jia Liu [1]

## Abstract

Collaborative training methods like Federated Learning (FL) and Split Learning (SL) enable distributed machine learning without sharing raw data. However, FL assumes clients can train entire models, which is infeasible for large-scale models. In contrast, while SL alleviates the client memory constraint in FL by offloading most training to the server, it increases network latency due to its sequential nature. Other methods address the conundrum by using local loss functions for parallel client-side training to improve efficiency, but they lack server feedback and potentially suffer poor accuracy. We propose FSL-SAGE (Federated Split Learning via Smashed Activation Gradient Estimation), a new federated split learning algorithm that estimates server-side gradient feedback via auxiliary models. These auxiliary models periodically adapt to emulate server behavior on local datasets. We show that FSL-SAGE achieves a convergence rate of $\mathcal{O}(1/\sqrt{T})$, where $T$ is the number of communication rounds. This result matches FedAvg, while significantly reducing communication costs and client memory requirements. Our empirical results also verify that it outperforms existing state-of-the-art FSL methods, offering both communication efficiency and accuracy.

## 1. Introduction

**1) Background and Motivation:** In the recent years, large foundation models (Devlin et al., 2019; Radford et al., 2021; Ramesh et al., 2022) and LLMs (OpenAI et al., 2024; Tou-

---

[1]Department of Electrical and Computer Engineering, The Ohio State University, Columbus, Ohio, USA [2]Department of Computer Science, University of Kentucky, Lexington, Kentucky, USA [3]Department of Industrial Engineering, The Ohio State University, Columbus, Ohio, USA [4]Air Force Research Laboratory, Rome, New York, USA. Correspondence to: Srijith Nair <nair.203@osu.edu>, Jia Liu <liu@ece.osu.edu>.

*Proceedings of the 42$^{nd}$ International Conference on Machine Learning*, Vancouver, Canada. PMLR 267, 2025. Copyright 2025 by the author(s).

vron et al., 2023) pretrained on large corpora of data have demonstrated near human-level performance on a variety of tasks. These models demand heavy computational resources and very large datasets for training (Kaplan et al., 2020; Hoffmann et al., 2022). The datasets are often owned by different organizations, making it impossible for a single entity to train a model on all the data (Sani et al., 2024b). Distributed learning paradigms like federated learning (FL) (McMahan et al., 2016) and split learning (SL) (Vepakomma et al., 2018), which preserve data privacy while leveraging the compute power of various clients, have become increasingly popular in LLM pretraining (Sani et al., 2024a) and fine-tuning (Sun et al., 2024; Wang et al., 2024).

Despite their benefits, FL and SL both have some limitations. Although FL can preserve data privacy by training the model on the client and aggregating the client-side models at a server, an inherent assumption of FL is that client devices have enough computational power to process the model, which is impractical for large models. SL, on the other hand, reduces compute requirements for clients by splitting the model into two parts, processing a smaller piece at the client, and offloading the other piece to the server. This, however, implies that processing the complete model in each iteration involves transmitting the cut-layer features and gradients back and forth between the client and the server. Moreover, SL loses the benefits of data parallelism in FL, since the server needs to assist gradient computation for each client in a sequential fashion. Consequently, SL suffers significant increases in communication load, power, and latencies.

To reduce the communication overheads incurred by SL, some recent works (Han et al., 2021; Mu & Shen, 2023) propose to use *auxiliary models* at the clients. These auxiliary models continue the forward propagation of the client-side model to compute a *local* loss function, which is then used to update both models via gradient descent. This eliminates the need for awaiting the server's response, which significantly speeds up the training process. Meanwhile, the server-side model is updated at a lower frequency by using the cut-layer features from the clients. While enjoying the benefit that the clients can conduct training in parallel, this can significantly reduce communication costs and latencies. However, the client-side models do not get any feedback from the server-side model, i.e., the clients are only updated

using the auxiliary models and thus never get trained via the server-side model.

To overcome these limitations, we propose a new approach called Federated Split Learning via Smashed Activation Gradient Estimation (FSL-SAGE). Note that the auxiliary model in other FSL frameworks can be interpreted as mimicking the server-side model at each client. Since, from the client-side model's perspective, the role of the server-side model is to provide the gradients of the loss function with respect to the cut-layer features, we train the auxiliary models to estimate these cut-layer gradients returned by the server-side model, which are then used to update the client-side models. This allows FSL-SAGE to continue to enjoy the benefits of FL and SL, while using the auxiliary models to accurately play the server's role. Also, we propose to update the auxiliary models periodically in a less frequent fashion, thus significantly reducing communication costs.

**2) Technical Challenges:** However, due to the following technical challenges, establishing finite-time convergence rate performance guarantee for FSL-SAGE is highly non-trivial and necessitates new proof and analysis techniques. First, as will be shown later in Fig. 1, there are multiple *coupled* time-scales involved during training, which include 1) the local step, whereby the client-side model is updated over a mini-batch of local data; 2) the lesser frequent server step, where the server-side model is updated given the cut-layer features; 3) a federated averaging step for the client-side models; and finally, 4) the least frequent *alignment* step, where the auxiliary models are updated to match the server-side model. Also, one needs to account for the auxiliary alignment step, which has *not* been studied in conventional SL methods (e.g., SplitFed (Thapa et al., 2022) or in the precursor to this work (Mu & Shen, 2023)). Exacerbating the problems in FSL-SAGE convergence rate analysis is the fact that the auxiliary models inject highly heterogeneous estimation errors during the training process, which are further coupled with all four time-scales.

**3) Main Contributions:** The key contribution of this paper is that we overcome the aforementioned technical challenges and rigorously establish the finite-time convergence rate guarantees of our proposed FSL-SAGE. Our main technical results are summarized as follows:

- We propose a new federated split learning algorithm called FSL-SAGE to train models that are too large to be trained on commodity client devices. By using a periodically aligned auxiliary model at each client to estimate the server-side model response, FSL-SAGE enables large model federated training, while enjoying low communication costs and the same data parallelism as in conventional FL.

- We establish a general finite-time stationary convergence rate bound for FSL-SAGE. Based on this general

result, we further show that under a mild in-expectation PAC-learnability assumption on the auxiliary models, FSL-SAGE can achieve an $\mathcal{O}(1/\sqrt{T})$ convergence rate, which is the same as those of state-of-the-art FL algorithms even with heterogeneous datasets. Our analysis sheds theoretical lights on the impacts of the class of auxiliary model functions on the final-time of convergence, which could be of independent interests to other FSL methods with auxiliary models.

- We further propose a "lazy version" of our FSL-SAGE method, where the auxiliary models are frozen beyond a certain point of alignment. This lazy version can be useful in practice to further reduce communication costs. We rigorously establish an explicit trade-off between this time and the accuracy of the final model.

- We conduct extensive experiments using large ResNet-18 and GPT2-medium models to verify our theoretical results and demonstrate that while being much more communication efficient than existing state-of-the-art FL/SL algorithms, the accuracy of FSL-SAGE is either on-par or even better.

## 2. Related Work

In this section, we provide a quick overview on several closely related areas, namely FL, SL, and FSL, thus putting our work in comparative perspectives.

**1) Federated Learning:** Federated learning was first proposed in McMahan et al. (2016) and theoretically analyzed in Yu et al. (2019), as an alternative to centralized learning, whereby several clients could collaborate on training a single ML model under the supervision of a server without the need for sharing client data. In FL, the model parameters are transmitted over the network to the client devices, where the clients can train the model on their local datasets before transmitting them back to the server where the models get aggregated. This process is repeated iteratively until the model converges. Federated learning enjoys the benefit that the model needs to be transmitted only once per communication round, and the clients can train their local models in parallel, for several iterations per round. Several FL methods (Yang et al., 2021; Karimireddy et al., 2019; Sahu et al., 2018; Reddi et al., 2021; Li & Lyu, 2023) have been analyzed in the recent years, most of which are variants of the original FedAvg algorithm (McMahan et al., 2016). As mentioned earlier, one of the key limitations of conventional FedAvg-type algorithms is that they assume each client has sufficient memory capacity to store the entire model to perform local gradient-based updates. However, this assumption becomes increasingly problematic as machine learning models continue to grow in size.

**2) Split Learning:** To alleviate the memory constraint in training large models with FL, an alternative strategy was

proposed in Vepakomma et al. (2018), where the idea is to split a large neural network model into two parts; the smaller part that contains the input layer is trained on the clients and the remaining larger portion is trained on the server. The layer at which the model is split is often called the *cut-layer*. During training, the cut-layer features for each mini-batch of data are communicated from the clients to the server, and the gradients of the loss with respect to (w.r.t.) the cut-layer features are in-turn relayed from the server back to the clients. Moreover, the server needs to sequentially process the server-side portion of the model for each client as discussed in Gupta & Raskar (2018), which renders high latency in using the vanilla SL approach. It is thus clear that SL reduces client memory requirements at the expense of slow and highly sequential training and increased communication costs. CPSL (Wu et al., 2023) is an extension of traditional SL, where clients are divided into clusters and clients within each cluster train in parallel. However, CPSL still suffers the same latencies as in SL.

**3) Federated Split Learning:** Since FL and SL each have inherent limitations, an important research is to explore whether we can design a new algorithmic framework that combines the data-parallel training efficiency of FL with the model-splitting capability of SL, thus enabling both faster training and reduced computational demands on clients. Toward this end, the SplitFed algorithm with two variants was proposed in Thapa et al. (2022), namely SplitFedv1 and SplitFedv2. SplitFedv1 maintains multiple copies of the server-side model at the server, one corresponding to each client, thereby allowing all clients to train in parallel, but at the expense of the non-scalable computational demand on the server. By contrast, in SplitFedv2, the server performs each client's backpropagation sequentially on a first-come-first-serve basis. The key advantage to both methods is that the clients can perform forward propagation in parallel. AdaptSFL (Lin et al., 2024) is a recent extension of SplitFedv2, where the authors adapted the client-side model size based on the available resources making SplitFed more flexible to resource-constrained clients. However, the communication overhead of these methods is the same as SL.

**4) Auxiliary Models:** The most related works to our paper are Han et al. (2021) and Mu & Shen (2023), where algorithms using *local cost functions* were proposed to avoid the need for the clients to wait for the server to return the cut-layer gradients. The client-side models are updated in parallel by backpropagating gradients via the local loss function through auxiliary models on the corresponding clients. The client-side and auxiliary models are then aggregated every round. The server-side model is updated by receiving cut-layer activations once in several local iterations. The method in Han et al. (2021) is similar to SplitFedv1 in the sense that separate server-side models are maintained per client, while the method in Mu & Shen

(2023) is similar to SplitFedv2, where only one server-side model is updated on a first-come-first-serve basis. However, our work differs from these existing methods in the following key aspect: the auxiliary models at the clients in our proposed FSL-SAGE are periodically aligned with the server-side model to ensure high training accuracy with low communication costs. In contrast, the auxiliary models in the aforementioned methods do not perform any alignments with the server-side model, thus leading to potentially low training accuracy. Notably, we circumvent all limitations of the above methods. Our method minimizes computational burden on the clients and communication costs that are inevitable in the other algorithms, and reduces latencies by facilitating parallel training, without compromising accuracy. Oh et al. (2022) addressed the inherent limitations of using local losses and the dual frequency of updates in FSL frameworks, and proposed to locally regularize the client-side models to maximize the mutual information between raw and cut-layer data, while also augmenting the smashed data so that the server-side model to improve latency and accuracy in parallel SL.

Group Knowledge Transfer (GKT) (He et al., 2020) is another approach related to our work. The basic idea of GKT is to train smaller client-side auxiliary models and then transfer their knowledge to the server via a variant of knowledge distillation (Hinton et al., 2015) by using the logits from the auxiliary model to train the server-side model. GKT is classified as a knowledge distillation approach with *modified* loss functions to incorporate server feedback during client-side model training, and is not exactly a federated split learning method. Thus, its performance is not directly comparable to FSL methods due to its different training objective function.

Although auxiliary models have been used in centralized deep network training Marquez et al. (2018); Bhatti & Moon (2022), the training of the auxiliary models in such works is indirect, and can be interpreted as training the network in parts. In our work, we explicitly train the auxiliary models to approximate the server-side models, thus ensuring their most accurate use.

## 3. The Proposed FSL-SAGE Algorithm

In this section, we present our system settings and the proposed FSL-SAGE algorithm.

**1) System Setting:** As shown in Fig. 1, our federated system consists of a server and $m$ clients indexed by $i$. Each client $i$ is associated with a local dataset $D_i$. The server contains two server processes: (i) the "$F$-server" for conducting federated aggregation, and (ii) the "$S$-server" that processes the server-side model and updates the auxiliary models for each client. In this paper, we call one cycle of federated

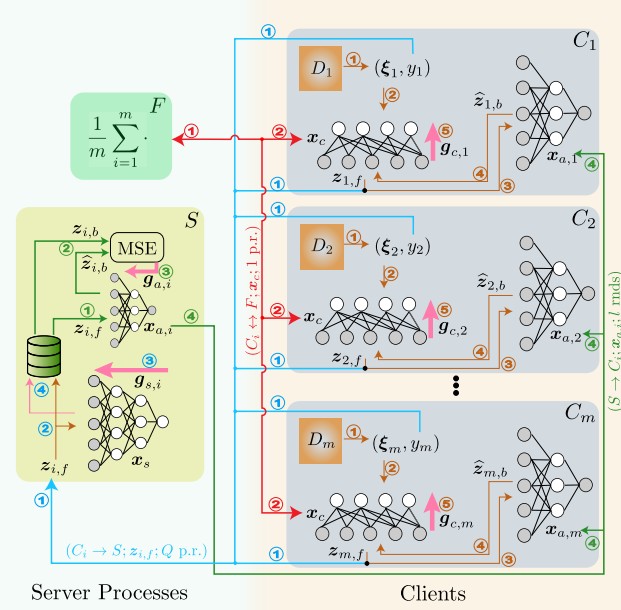

*Figure 1.* Schematic diagram of the FSL-SAGE algorithm. Text on the arrows indicates (sender → receiver, message, rate), e.g., the red arrows transmit $\boldsymbol{x}_c$ between $C_i$ and the $F$-server at the rate of 1 per round (p.r.), the updated $\boldsymbol{x}_{a,i}$ is sent to $C_i$ once in $l$ rounds. Arrows are color coded: brown arrows represent local operations, blue arrows represent the smashed data and label transmission to the server, red arrows indicate federated averaging, and green arrows indicate auxiliary model transfer; circled numbers indicate the order of operations

aggregation as one "round" and assume that FSL-SAGE runs for a total of $T$ rounds. Our objective is to learn a model $\boldsymbol{x} \in \mathbb{R}^d$ that minimizes a global loss function $f : \mathbb{R}^d \to \mathbb{R}$:

$$\min_{\boldsymbol{x} \in \mathbb{R}^d} f(\boldsymbol{x}) := \frac{1}{m} \sum_{i=1}^m F_i(\boldsymbol{x}), \qquad (1)$$

where $F_i(\cdot) := \mathbb{E}_{\boldsymbol{\xi}_i \sim D_i}[F_i(\boldsymbol{x}; \boldsymbol{\xi}_i)]$ corresponds to the objective function of the $i$-th client.

**2) The FSL-SAGE Algorithm:** The entire process of FSL-SAGE is illustrated in Fig. 1. Similar to split learning, in FSL-SAGE, we split the model $\boldsymbol{x}$ into the client-side model $\boldsymbol{x}_{c,i}$ for all clients $i \in [m]$ and the server-side model $\boldsymbol{x}_s$. As shown in Fig. 1, a single local iteration at the $i$-th client consists of the following key steps: (1) The client-side model takes as input a sampled batch of data $\boldsymbol{\xi}_i \in D_i$ and outputs the cut-layer features $\boldsymbol{z}_{i,f}(\boldsymbol{x}_{c,i}; \boldsymbol{\xi}_i)$, where the subscript $f$ in $\boldsymbol{z}_{i,f}$ denotes the output of the "forward" pass; (2) Each client has an auxiliary model $\boldsymbol{x}_{a,i}$, which takes as input $\boldsymbol{z}_{i,f}$ and computes the cost function estimate $\widehat{F}(\boldsymbol{x}_{a,i}; \boldsymbol{z}_{i,f})$; (3) In the backward pass, the auxiliary model computes the gradients of the cost estimate with respect to the cut layer features, denoted as $\widehat{\boldsymbol{z}}_{i,b}(\boldsymbol{x}_{a,i}; \boldsymbol{z}_{i,f}, y_i)$, where $y_i$ is the label corresponding to $\xi_i$, and the subscript $b$ in $\widehat{\boldsymbol{z}}_{i,b}$ denotes the result of the 'backward' pass. These gradients

---

**Algorithm 1** The Code of Client $i$ in FSL-SAGE.

**Require:** $\boldsymbol{x}_{a,i}^0$
1: **for** $t \leftarrow 0, 1, \dots, (T-1)$ **do**
2:     $\boldsymbol{x}_{c,i}^{t,0} \leftarrow \boldsymbol{x}_c^t$        # Initialize weights from F-server
3:     **if** $t \equiv 0 \mod l$ **then**
4:         **Receive** $\boldsymbol{x}_{a,i}^t$ from $S$-server
5:     **else**
6:         $\boldsymbol{x}_{a,i}^t \leftarrow \boldsymbol{x}_{a,i}^{t-1}$
7:     **end if**
8:     **for** $k \leftarrow 1, 2, \dots, (K-1)$ **do**
9:         $(\boldsymbol{\xi}_i^{t,k}, y_i^{t,k}) \sim D_i$     # Sample local mini-batch
10:       $\boldsymbol{z}_{i,f}^{t,k} \leftarrow \boldsymbol{z}_{i,f}(\boldsymbol{x}_{c,i}^{t,k}; \boldsymbol{\xi}_i^{t,k})$  # Forward pass on client
11:       **if** $k \equiv 0 \mod (K/Q)$ **then**
12:           **Send** $(\boldsymbol{z}_{i,f}^{t,k}, y_i^{t,k})$ to $S$-server
13:       **end if**
14:       $\widehat{\boldsymbol{z}}_{i,b}^{t,k} \leftarrow \widehat{\boldsymbol{z}}_{i,b}(\boldsymbol{x}_{a,i}^t; \boldsymbol{z}_{i,f}^{t,k})$  # Compute grad estimate
15:       $\boldsymbol{x}_{c,i}^{t,k+1} \leftarrow \boldsymbol{x}_{c,i}^{t,k} - \eta_L \boldsymbol{J}_{c,i}^{t,k} \widehat{\boldsymbol{z}}_{i,b}^{t,k}$  # Update client
16:     **end for**
17:     **Send** $x_{c,i}^{t,K}$ to $F$-server
18: **end for**

---

**Algorithm 2** FSL-SAGE $F$-Server

1: Initialize model $\boldsymbol{x}^0$
2: $(\boldsymbol{x}_c^0, \boldsymbol{x}_s^0) \leftarrow$ **split** $(\boldsymbol{x}^0)$       # Split model
3: **Send** $\boldsymbol{x}_s^0$ to Server $S$
4: **Broadcast** $\boldsymbol{x}_c^0$ to all clients
5: **for** $t \leftarrow 0, 1, \dots, T-1$ **do**
6:     **Receive** $\boldsymbol{x}_{c,i}^{t,K}$ from clients $i = 1, 2, \dots, m$
7:     $\boldsymbol{x}_c^{t+1} = \frac{1}{m} \sum_{i=1}^m \boldsymbol{x}_{c,i}^{t,K}$   # Aggregate client model
8:     **Broadcast** $\boldsymbol{x}_c^{t+1}$ to all clients
9: **end for**

---

are in turn used to update the client-side model; (4) Every client performs $K$ local iterations before sending their models, $\boldsymbol{x}_{c,i}$, to the $F$-server for aggregation. Algorithms 1 and 2 summarize the client and $F$-server operations in FSL-SAGE.

To update the server-side model, the clients periodically send the computed cut-layer features and labels to the $S$-server $Q$ times per round. The $S$-server computes the true loss function $F_i(\boldsymbol{x}_s; \boldsymbol{z}_{i,f}, y_i)$ and the gradient of $F_i$ with respect to $\boldsymbol{z}_{i,f}$, which is denoted by $\boldsymbol{z}_{i,b}$. The server-side model is then updated in a gradient descent fashion. The tuple $(\boldsymbol{z}_{i,f}, y_i)$, called the *alignment dataset*, is stored for training the auxiliary model. Algorithm 3 summarizes the operations of the $S$-server.

Once every $l$ rounds, the server initiates an *alignment process*, where each client's auxiliary model is updated using the respective alignment dataset. Assuming round $t$ is a multiple of $l$ the auxiliary model at round $t$, is updated as

**Algorithm 3** (Lazy) FSL-SAGE $S$-Server

1:  **Receive** $x_s^0$ from $F$-server
2:  **for** $t \in [T]$ **do**
3:      **for** $k \in [K]$ **if** $k \equiv 0 \mod K/Q$ **do**
4:          **Receive** $(z_{i,f}^{t,k}, y_i) \ \forall i$ on FCFS basis
5:          **Compute** grad $g_s(x_s^{t,k}; z_{i,f}^{t,k}, y_i)$ via backprop
6:          $x_s^{t,k+1} := x_s^{t,k} - \eta g_s^{t,k}$     *# Update server model*
7:          **Store** for client $i$: $(z_{i,f}^j, y_i^j)$
8:      **end for**
9:      **if** $t \equiv 0 \mod l$ **and** $t \leq T'$ **then**     *# $T' \leq T$*
10:         **for** client $i \leftarrow 1, 2, \ldots, m$ **do**
11:             **Compute** $z_{i,b}(z_{i,f}^j, y_i^j) \ \forall j$ from
12:             $x_{a,i}^t = \mathbf{align}\left( \left\{ z_{i,f}^j, z_{i,b}^j, y_i^j \right\}_j \right)$
13:             **Send** $x_{a,i}^t$ to client $i$
14:         **end for**
15:     **end if**
16: **end for**

the following:

$$x_{a,i}^t = \arg\min_{x_a} \frac{1}{2} \sum_{j=1}^{Qt} \left\| \widehat{z}_{i,b}(x_a; z_{i,f}^j, y_i^j) - z_{i,b}^j \right\|_2^2, \quad (2)$$

where the backward gradients $z_{i,b}^j$ are computed by passing $z_{i,f}^j$ and $y_i^j$ through the server-side model. In practice, the minimization in (2) could be approximately done by using a finite number of steps of gradient descent, Adam (Kingma & Ba, 2015), or other optimizers. Also, since the alignment dataset grows larger in size with the number of rounds increases, one can discard the older entries in order to maintain a fixed maximum size of the dataset. For simplicity in theoretical analysis, we assume that the server has sufficiently large storage to accommodate the growing alignment dataset size.

In Algorithm 1, if the auxiliary models are only updated for the first $T' < T$ rounds of communication and then are frozen until the end of training, we will refer to this version as the Lazy FSL-SAGE algorithm. Note that, in practice, choosing a smaller $T'$ can further reduce communication costs at the expense of final accuracy of the learned model. On the other hand, the FSL-SAGE algorithm is non-lazy if $T' = T$. In Section 4, we will characterize the stationarity gaps of both lazy and non-lazy versions of FSL-SAGE.

## 4. Theoretical Convergence Analysis

For convenient reference, we summarize the key notation used in the following analysis in Appendix A and Table 2. We begin by making the following standard assumptions widely adopted in the FL literature (Karimireddy et al., 2019;

Reddi et al., 2021; Yang et al., 2021).

**Assumption 4.1** (Smoothness). For all $i = 1, 2, \ldots m$, the following objective functions corresponding to the $i$-th client are smooth, i.e., for all $x_s, x_{a,i}, \xi_i$ and $y_i$:

(a) The gradient of $g_i := \nabla F_i(x_c, x_s; \xi_i, y_i)$ is $L_c$-Lipschitz with respect to $x := (x_c, x_s)$, i.e., there exists a constant $L_c > 0$ such that $\|g_i(u_c, u_s; \xi_i, y_i) - g_i(v_c, v_s; \xi_i, y_i)\| \leq L_c \|u - v\|$, for all $u := (u_c, u_s)$ and $v := (v_c, v_s) \in \mathcal{X}$.

(b) The gradient of the local client objective $\widehat{F}_i$, $\widehat{g}_{c,i} := \nabla_c \widehat{F}_i(x_c, x_{a,i}; \xi_i, y_i)$ is $\widehat{L}_c$-Lipschitz with respect to $x_c$: there exists a constant $\widehat{L}_c > 0$ such that $\|\widehat{g}_{c,i}(u_c, x_{a,i}; \xi_i, y_i) - \widehat{g}_{c,i}(v_c, x_{a,i}; \xi_i, y_i)\| \leq \widehat{L}_c \|u_c - v_c\|$ for all $u_c, v_c \in \mathcal{X}_c$.

**Assumption 4.2** (Bounded Variance). The following quantities measuring variability of the local stochastic estimates of the objective function and the variability of the client loss function are bounded:

(a) The local gradient estimate for a mini-batch sampled at client $i$ is unbiased for all $i$, i.e., $\forall \ \widetilde{x}, \ \mathbb{E}_{\xi_i \sim \mathcal{D}_i}\left[ \nabla \widehat{F}_i(\widetilde{x}; \xi_i) \right] = \nabla \widehat{F}_i(\widetilde{x}_i)$ and has a bounded variance, i.e., $\exists \ \sigma_L > 0$ such that $\mathbb{E}_{\xi_i \sim \mathcal{D}_i}\left[ \left\| \nabla \widehat{F}_i(\widetilde{x}; \xi_i) - \nabla \widehat{F}_i(\widetilde{x}) \right\|^2 \right] \leq \widehat{\sigma}_L^2$;

(b) The local gradient for client $i$ is unbiased for all $i$, i.e., $\forall \ \widetilde{x}, \ \mathbb{E}_{\xi_i \sim \mathcal{D}_i}\left[ \nabla F_i(x; \xi_i) \right] = \nabla F_i(x_i)$ and has a bounded variance, i.e., $\exists \ \sigma_L > 0$ such that $\mathbb{E}_{\xi_i \sim \mathcal{D}_i}\left[ \|\nabla F_i(x; \xi_i) - \nabla F_i(x)\|^2 \right] \leq \sigma_L^2$

(c) The global variability of the local client gradient is bounded, i.e., for all $x$ and $i \in [m]$, $\|\nabla F_i(x) - \nabla f(x)\|^2 \leq \sigma_G^2$.

Assumptions 4.1 and 4.2 are commonly used in analyzing convergence of many optimization algorithms from the classical SGD (Ghadimi & Lan, 2013) to many variants of federated learning (Yu et al., 2019; Yang et al., 2021). The variance terms $\widehat{\sigma}_L^2, \sigma_L^2$ and $\sigma_G^2$ in Assumption 4.2 characterize the degree of intra-client and inter-client data heterogeneity.

### 4.1. General Finite-Time Convergence Rate Results of Non-lazy FSL-SAGE

In what follows, we provide an upper bound on the *stationarity gap* of the FSL algorithm, i.e., a bound on the expected squared norm of the gradient of the model parameters.

**Theorem 4.3** (Convergence of Non-Lazy FSL-SAGE). *Under Assumptions 4.1 and 4.2, given step-sizes $\eta_L$ and $\eta$*

*satisfying $\eta \leq \frac{1}{4\sqrt{2}QL_c}$ and $\eta_L \leq \frac{1}{2\sqrt{10}K\widehat{L}_c}$, FSL-SAGE, as described in Algorithms 1-3, has the following finite-time stationarity convergence rate bound:*

$$\min_{t \in [T]} \mathbb{E}\left[\left\|\nabla f(\boldsymbol{x}^t)\right\|^2\right] \leq \frac{f(\boldsymbol{x}^0) - f^*}{c \min\{\eta_L, m\eta\}QT}$$

$$+ \frac{3K\eta_L}{2cQ \min\{\eta_L, m\eta\}} \frac{1}{T}\sum_{t=1}^{T} \varepsilon^t + \frac{\Phi(\eta_L, \eta)}{T}, \quad (3)$$

*where*

$$\Phi(\eta_L, \eta) := \frac{c'K}{2c^2 mQ \min\{\eta_L, m\eta\}} \cdot \big[\eta_L^2 \widehat{\sigma}_L^2 L_c$$

$$+ 40mK\eta_L^3 \widehat{L}_c^2(\widehat{\sigma}_L^2 + K\sigma_G^2)$$

$$+ m^2\eta^2\sigma_L^2 L_c + 32Qm^2\eta^3 L_c^2(\sigma_L^2 + Q\sigma_G^2)\big]. \quad (4)$$

*and $\varepsilon^t$ is the gradient estimation error incurred by the auxiliary model averaged over clients $i \in [m]$:*

$$\varepsilon^t := \frac{1}{m}\sum_{i=1}^{m} \left\|\nabla \widehat{F}_i(\widetilde{\boldsymbol{x}}^t) - \nabla F_i(\boldsymbol{x}^t)\right\|^2. \quad (5)$$

A full proof of Theorem 4.3 is provided in Appendix B.1. The following result directly follows from Theorem 4.3.

**Corollary 4.4.** *For the step-size choices $\eta = \mathcal{O}\left(\frac{1}{m\sqrt{T}}\right)$ and $\eta_L = \mathcal{O}\left(\frac{1}{\sqrt{T}}\right)$, Algorithms 1-3 achieves the following finite-time stationarity convergence rate:*

$$\min_{t \in [T]} \mathbb{E}\left[\left\|\nabla f(\boldsymbol{x}^t)\right\|^2\right] = \mathcal{O}\left(\frac{1}{\sqrt{T}}\right) + \frac{\mathcal{O}(1)}{T}\sum_{t=1}^{T} \varepsilon^t. \quad (6)$$

The last term in (6) corresponds to the round-average of the auxiliary models' estimation error in estimating the cut-layer gradients returned by the server, which plays an important role in the convergence of our method. While these terms similar to terms 1 and 3 in the RHS of (3) appear in convergence proofs of FedAvg (McMahan et al., 2016) and its variants (Yang et al., 2021; Reddi et al., 2021), the estimation error term is a *direct result of using our auxiliary model* to estimate the server gradients. We note that, although CSE-FSL (Mu & Shen, 2023) also uses auxiliary models, their analysis only considers the client-side and server-side models separately, and thus they bypass analyzing such an important estimation error term.

## 4.2. $\mathcal{O}(\frac{1}{\sqrt{T}})$ Convergence Rate of Non-lazy FSL-SAGE with Agnostic PAC Learnable Auxiliary Models

With the general finite-time convergence rate results in Theorem 4.3, it remains unclear whether one can achieve the same $\mathcal{O}(\frac{1}{\sqrt{T}})$ convergence rate as the conventional FedAvg-type methods. In this section, we show that the answer is

"yes" if we further impose a mild learnability assumption on the auxiliary model. Toward this end, we first introduce the notion called in-expectation learnability (Mey, 2022):

**Definition 4.5** (In-Expectation Learnability). The hypothesis class $\mathcal{G}$ is said to be *in-expectation learnable* by the empirical risk minimization (ERM) algorithm if and only if $\forall \epsilon > 0$, there exists a $r_{\mathcal{G}}(\epsilon)$ such that, for $r \geq r_{\mathcal{G}}(\epsilon)$ training samples, the following bound holds:

$$\mathbb{E}\left[\left\|\boldsymbol{g}(\boldsymbol{\theta}_r; \boldsymbol{x}) - \boldsymbol{f}(\boldsymbol{x})\right\|^2\right]$$

$$\leq \min_{\boldsymbol{\theta}} \mathbb{E}\left[\left\|\boldsymbol{g}(\boldsymbol{\theta}; \boldsymbol{x}) - \boldsymbol{f}(\boldsymbol{x})\right\|^2\right] + \epsilon, \quad (7)$$

where the expectation is over $\boldsymbol{x}, \boldsymbol{x}_1, \boldsymbol{x}_2, \ldots, \boldsymbol{x}_r$; and $\boldsymbol{\theta}_r$ is the ERM hypothesis learned from $\{\boldsymbol{x}_i\}_{i=1}^r$ that are i.i.d sampled from $\mathcal{D}$:

$$\boldsymbol{\theta}_r(\{\boldsymbol{x}_i\}_i) := \arg\min_{\boldsymbol{\theta} \in \Theta} \frac{1}{r}\sum_{i=1}^{r} \left\|\boldsymbol{g}(\boldsymbol{\theta}; \boldsymbol{x}_i) - \boldsymbol{f}(\boldsymbol{x}_i)\right\|^2. \quad (8)$$

Next, we introduce additional notation pertaining to the loss function used to train the auxiliary model. We start by denoting the hypothesis class of auxiliary model functions for the $i^{\text{th}}$ client as $\mathcal{A}_i := \{\widehat{\boldsymbol{z}}_{i,b}(\boldsymbol{x}_{a,i}; \cdot) : \boldsymbol{x}_{a,i} \in \mathcal{X}_a\}$, which is parameterized by $\boldsymbol{x}_{a,i}$. Omitting obvious parameters for readability, we can denote the empirical loss on $r$ training samples $\{(\boldsymbol{z}_{i,f}^j, y_i^j)\}_{j=1}^r$ as

$$\widehat{\mathcal{L}}_i(\boldsymbol{x}_{a,i}, \boldsymbol{x}) := \frac{1}{r}\sum_{j=1}^{r} \left\|\widehat{\boldsymbol{z}}_{i,b}(\boldsymbol{x}_{a,i}) - \boldsymbol{z}_{i,b}(\boldsymbol{x})\right\|^2, \quad (9)$$

and the true loss as

$$\mathcal{L}_i(\boldsymbol{x}_{a,i}, \boldsymbol{x}) := \mathbb{E}_{(\xi,y)\sim\mathcal{D}_i}\left[\left\|\widehat{\boldsymbol{z}}_{i,b}(\boldsymbol{x}_{a,i}) - \boldsymbol{z}_{i,b}(\boldsymbol{x})\right\|^2\right]. \quad (10)$$

Recall that the auxiliary model is trained once every $l$ rounds on an increasing number of smashed data samples $r := \lfloor t/l \rfloor$. Thus, one can define the ERM solution at round $t$ from Algorithms 1 & 3, and the corresponding optimal solution as follows:

$$\boldsymbol{x}_{a,i}^t = \arg\min_{\boldsymbol{x}_a \in \mathcal{X}_a} \widehat{\mathcal{L}}_i(\boldsymbol{x}_a, \boldsymbol{x}^t) \quad (11)$$

$$\boldsymbol{x}_{a,i}^{t\star} = \arg\min_{\boldsymbol{x}_a \in \mathcal{X}_a} \mathcal{L}_i(\boldsymbol{x}_a, \boldsymbol{x}^t). \quad (12)$$

Then, the auxiliary model (11) is an in-expectation learner of $\mathcal{A}_i$ if $\forall \epsilon > 0$, there exists a $r_i(\epsilon)$ such that, for $r \geq r_i(\epsilon)$:

$$\mathbb{E}\left[\mathcal{L}_i(\boldsymbol{x}_{a,i}^t, \boldsymbol{x}^t)\right] \leq \mathcal{L}_i(\boldsymbol{x}_{a,i}^{t\star}, \boldsymbol{x}^t) + \epsilon \quad (13)$$

This leads us to introduce the following learnability assumption on the auxiliary model.

**Assumption 4.6** (Learnable Auxiliary Model). The auxiliary model of the $i$-th client $\boldsymbol{x}_{a,i}$ in (9) is an in-expectation learner of $\mathcal{A}_i$ according to Definition 4.5 with sample complexity $r_i(\epsilon) = \mathcal{O}\left(1/\epsilon^2\right)$.

In (Grunwald et al., 2021), in-expectation bounds under the PAC-Bayesian formulation have been derived for loss functions satisfying the so-called $(B, \beta)$-Bernstein condition (Bartlett & Mendelson, 2006) with $B > 0, \beta \in [0, 1]$. The sample complexity of those bounds are given by $\mathcal{O}\left(\left\{\frac{comp}{r}\right\}^{\frac{1}{2-\beta}}\right)$ where $comp$ is some complexity measure that appears in the form of KL-divergence (Audibert, 2004; Zhang, 2006) or the conditional mutual information (CMI) (Steinke & Zakynthinou, 2020) of the posterior and prior distributions in PAC-Bayesian framework. Provided the $comp$ term does not increase drastically with $r$, Assumption 4.6 can be satisfied by such models. Lastly, we make the following Lipchitz client assumption:

**Assumption 4.7** (Lipschitz client). For all clients $i = 1, 2, \ldots m$ and for all $\boldsymbol{u}_c, \boldsymbol{v}_c \in \mathcal{X}_c$, the client-side cut-layer activation $\boldsymbol{z}_{i,f}(\boldsymbol{x}_c; \boldsymbol{\xi}_i)$ is $L_{zf}$-Lipschitz with respect to the client parameters $\boldsymbol{x}_c$, i.e., there exists a constant $L_{zf} > 0$ such that $\|\boldsymbol{z}_{i,f}(\boldsymbol{u}_c; \boldsymbol{\xi}_i) - \boldsymbol{z}_{i,f}(\boldsymbol{v}_c; \boldsymbol{\xi}_i)\| \leq L_{zf} \|\boldsymbol{u}_c - \boldsymbol{v}_c\|$.

With these assumptions, we are now in a position to state our next main result:

**Theorem 4.8** (FSL-SAGE with PAC Auxiliary Models). *Under Assumptions 4.1, 4.2, and 4.7, and with step-sizes $(\eta, \eta_L)$ satisfying the conditions in Theorem 4.3, if the auxiliary model satisfies Assumption 4.6 with $r_i(\epsilon) = \mathcal{O}\left(1/\epsilon^2\right)$, then after $T$ rounds, the iterates in FSL-SAGE satisfy:*

$$\min_{n \in \{1, \ldots, \lfloor T/l \rfloor\}} \mathbb{E}\left[\left\|\nabla f(\boldsymbol{x}^{nl-1})\right\|^2\right] \leq \frac{f(\boldsymbol{x}_0) - f^*}{c \min\{\eta_L, m\eta\} QT}$$
$$+ \frac{3CK\eta_L}{2Q \min\{\eta_L, m\eta\}\sqrt{T}} + \frac{\Phi(\eta_L, \eta)}{T}$$
$$+ \frac{3K\eta_L L_f^2}{2cQ \min\{\eta_L, m\eta\}} \frac{1}{T} \sum_{i=1}^{T} \varepsilon_\star^t \qquad (14)$$

*where $C > 0$ and $0 < c < 0.5 - 20K^2 \eta_L^2 \widehat{L}_c^2$ are some constants, and $\varepsilon_\star^t := \frac{1}{m} \sum_{i=1}^{m} \mathcal{L}_i(\boldsymbol{x}_{a,i}^{t\star}, \boldsymbol{x}^t)$.*

Note that, for the step-size choices $\eta_L = \mathcal{O}(1/\sqrt{T})$ and $\eta = \mathcal{O}(1/(m\sqrt{T}))$, the first three terms in the upper bound in (14), decrease with rounds as $\mathcal{O}(1/\sqrt{T})$. The last term, given by $C/T \sum_{t=1}^{T} \varepsilon_\star^t$ where $C = \mathcal{O}(1)$, is the average of $\varepsilon_\star^t$ over all rounds. The term $\varepsilon_\star^t$ is the error achieved by the best hypothesis in $\mathcal{A}$ at round $t$ and is entirely determined by the architecture of the auxiliary model. It is trivially zero for cases when the server-side model can be obtained for some realization of the auxiliary model parameters. A more interesting case is when the auxiliary model is typically much smaller than the server-side model. For 2-layer auxiliary

models, there are several results such as the *universal approximation theorem* (Cybenko, 1989; Hornik et al., 1989), and bounds for sufficiently wide or deep networks (Lu et al., 2017; Hanin & Sellke, 2018; Hanin, 2019) that show that $\varepsilon_\star^t$ can be made arbitrarily small. Finally, from the above discussions, we have the following complexity results:

**Corollary 4.9.** *For the step-size choices $\eta_L = \mathcal{O}(1/\sqrt{T})$ and $\eta = \mathcal{O}(1/(m\sqrt{T}))$, the non-lazy FSL-SAGE with PAC-learnable auxiliary models achieves a finite-time convergence rate of $\mathcal{O}(1/\sqrt{T})$.*

### 4.3. Convergence Results of Lazy FSL-SAGE

Since the alignment process is expensive in terms of communication costs, in practice one may want to stop performing alignment of the auxiliary model after $T' < T$ rounds, i.e., the Lazy FSL-SAGE apporach. Since the only difference in the convergence bound of non-lazy FSL-SAGE in Theorem 4.8 and Lazy FSL stems from the estimation error's contribution to the upper bound $\varepsilon^t$, we immediately have the following result for the Lazy FSL-SAGE with slight modification of the bound in Theorem 4.8:

**Corollary 4.10** (Convergence Rate of Lazy FSL). *Under Assumptions 4.1, 4.2, 4.6 and 4.7, and the step-sizes $(\eta, \eta_L)$ satisfying the conditions in Theorem 4.3, let us additionally assume that $\mathcal{A}$ is in-expectation learnable as per Definition 4.5 with $r_i(\epsilon) = \mathcal{O}\left(1/\epsilon^2\right)$ for all $\delta$, then after $T > T'$ rounds Lazy FSL-SAGE satisfies:*

$$\min_{n \in \{1, \ldots, \lfloor T/l \rfloor\}} \mathbb{E}\left[\left\|\nabla f(\boldsymbol{x}^{nl-1})\right\|^2\right] \leq \frac{f(\boldsymbol{x}_0) - f^*}{c \min\{\eta_L, m\eta\} QT}$$
$$+ \frac{3CK\eta_L(1 + T'/T)}{2Q \min\{\eta_L, m\eta\}\sqrt{T'}} + \frac{\Phi(\eta_L, \eta)}{T}$$
$$+ \frac{3K\eta_L L_f^2}{2cQ \min\{\eta_L, m\eta\}} \frac{1}{T} \sum_{i=1}^{T} \varepsilon_\star^t \qquad (15)$$

*where $c, C$ and $\varepsilon_\star^t$ are as defined in Theorem 4.8.*

Although Han et al. (2024) studied the convergence of SplitFed without any auxiliary models or local losses, to add to the challenges faced in their setting, in our case, the client-side model is updated through an *approximation* of the server-side model, which introduces approximation errors in the client-side model. To the best of our knowledge, our work is the first to provide an explicit relationship between the convergence rate and the approximation power of the auxiliary models.

An important limitation between FSL with sequentially processed server-side model and FL, which manifests in our convergence analysis is the *lack of linear speedup* (Yang et al., 2021), i.e., the convergence rate does not reduce with the increase in the number of clients $m$. Note that in all FSL approaches (Han et al., 2024; Mu & Shen, 2023), including

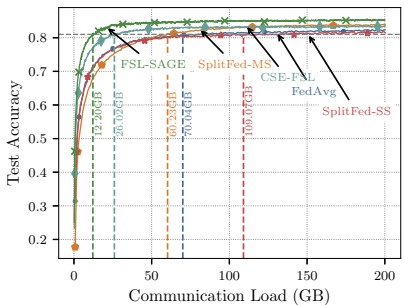 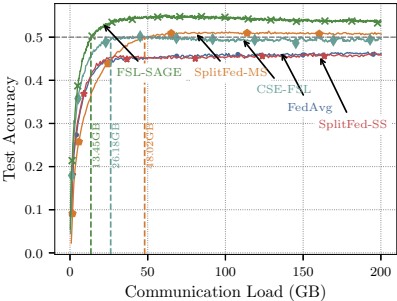 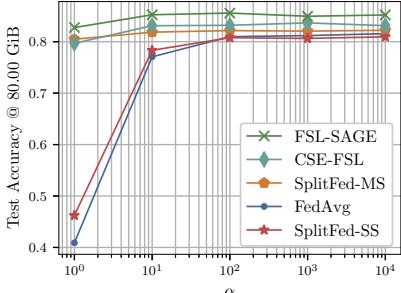

*Figure 2.* Test accuracy vs. communication load for ResNet-18 on CIFAR-10 distributed homogeneously across 10 clients. Curves are labeled from best (leftmost) to worst (rightmost) in final accuracy.

*Figure 3.* Test accuracy vs. communication load for ResNet-18 on CIFAR-100 distributed homogeneously across 10 clients.

*Figure 4.* Best accuracy vs. Dirichlet $\alpha \in (0, \infty]$ for ResNet-18/CIFAR-10 with non-i.i.d. data for 10 clients up to 80 GiB comm.

in our work, where a single server-side model is maintained, the server-side model must be sequentially processed, thus losing linear speedup.

## 5. Experimental Results and Discussion

In this section, we conduct numerical experiments to verify the efficacy of our proposed FSL-SAGE algorithm[1].

**1) Experiment Settings:** *1-a) Compute and Baselines:* We compare FSL-SAGE with FedAvg (McMahan et al., 2016), SplitFedv1 and SplitFedv2 (Thapa et al., 2022), and CSE-FSL (Mu & Shen, 2023). We use PyTorch for training on a single NVIDIA H100 NVL GPU with 80GB of memory.

*1-b) Datasets:* Although FL generally applies to a wide range of machine learning tasks, we focus on two tasks: 1) image classification on CIFAR-10 and CIFAR-100 datasets (Krizhevsky et al., 2009); and 2) natural language generation using the E2E (Novikova et al., 2017) dataset. The CIFAR datasets contain 60K $32 \times 32$ 3-channel images with 10 and 100 classes respectively. To simulate the effect of data heterogeneity we use the Dirichlet distribution parameterized by $\alpha \in (0, \infty)$ to determine the proportion of class labels (Hsu et al., 2019), where smaller $\alpha$ indicates more heterogeneity of class label distribution.

*1-c) Models:* For image classification, we use the ResNet-18 (He et al., 2016), which comprises of 4 ResNet blocks. We split the network in between block 2 and 3 to create a client-side model with 685K parameters and a server-side model with 10.5M parameters. For the auxiliary models, we arbitrarily cascade the first server ResNet block with the final fully-connected layer, yielding 2.1M parameters. For natural language generation, we perform LoRA finetuning (Hu et al., 2021) of GPT2-medium (Radford et al., 2019). We split the model after the first attention block, yielding

66.2M parameters for the client-side model and 365.7M parameters for the server-side model. For the auxiliary model, we use the first 3 attention blocks cascaded with the language decoder of the server-side model (92.4M params).

*1-d) Hyperparameters:* For image classification, we use a batch-size of 256. Clients train their models for 1 epoch on their local dataset per federated averaging round. For CSE-FSL and FSL-SAGE, the cut-layer features are sent to the server-side model every 5 local steps, and for FSL-SAGE, the auxiliary models are aligned with the server every $l = 10$ rounds. We stop training when the communication cost incurred exceeds 200 GiB. The client-side and server-side models are optimized using Adam (Kingma & Ba, 2015) with a learning rate $10^{-3}$, weight decay $10^{-4}$, and $\beta_1 = 0.9, \beta_2 = 0.999$. For alignment, we use the same optimizer settings with no weight decay. For better interpretability, we use SplitFed-MS (multi-server) in lieu of SplitFedv1 and SplitFed-SS (single-server) in lieu of SplitFedv2.

**2) Results and Discussions:** *2-a) Image Classification:* In Fig. 2, we plot the test accuracy against communication load on 10 clients on CIFAR-10, averaged over 4 training runs. FSL-SAGE outperforms all the other methods in final accuracy. More importantly, for a given level of accuracy, say 81%, FSL-SAGE achieves almost $2.2\times$ lesser communication cost than CSE-FSL, and $10\times$ lesser than SplitFed-SS. Figure 3 depicts a similar plot for CIFAR-100. Table 1 summarizes the best accuracies on CIFAR datasets.

In Fig. 4, we plot the best accuracy achieved by each method

| Algorithm | CIFAR-10 | | CIFAR-100 | |
|---|---|---|---|---|
| | iid | non-iid | iid | non-iid |
| FedAvg | 82.08 | 42.72 | 45.79 | 35.26 |
| SplitFedv1 | **84.21** | **81.48** | 50.78 | **51.03** |
| SplitFedv2 | 81.42 | 46.79 | 45.54 | 35.57 |
| CSE-FSL | 83.74 | 80.40 | **50.94** | 49.55 |
| FSL-SAGE | **85.71** | **82.75** | **56.06** | **51.90** |

*Table 1.* Best test accuracy (%) for ResNet-18 trained up to 200 GiB. Best and second best results are colored and bold.

[1] Our source code is available at https://github.com/srijith1996/FSL-SAGE.

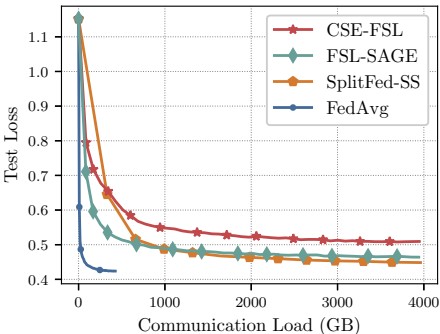

*Figure 5.* Test loss vs. communication load for LoRA finetuning of GPT2-m on i.i.d. E2E distributed across 3 clients.

over the course of training on CIFAR-10, upto a communication load of $80\,\mathrm{GiB}$, against the Dirichlet distribution parameter $\alpha \in (0, \infty)$. We vary $\alpha$ from $10^{-1}$ (high heterogeneity) to $10^4$ (low heterogeneity) to simulate the effect of uneven data distribution among 10 clients. While FedAvg and SplitFed-SS do not converge within the specified communication budget under high heterogeneity, the other three methods are much more robust to data heterogeneity. FSL-SAGE is robust within the given range of heterogeneity, and outperforms other methods. We present some supplementary plots for the image classification experiments in Appendix D.

*2-b) LoRA Fine-tuning GPT2 on E2E:* In order to study the convergence rates of different FL/SL methods, we also perform an experiment on fine-tuning the pretrained GPT2-medium (Radford et al., 2019) on the E2E dataset (Novikova et al., 2017), which is a tabular to natural language generation problem. In Fig. 5, we plot the masked cross-entropy loss against communication load for test data. The methods are run upto $4\,\mathrm{TB}$ of communication or $50$ rounds, whichever is earlier. We observe that FSL-SAGE converges faster, and is more accurate than its auxiliary-based counterpart CSE-FSL, and almost as fast as SplitFed-SS. FedAvg converges the fastest since, for GPT2-medium, communicating the model once per round consumes much lesser bytes than more frequently transmitting the smashed data.

## 6. Conclusion

In this paper, we proposed a new federated split learning algorithm, FSL-SAGE, which facilitates the training of large models using FL, while enjoying the benefits of data parallelism. Our method leverages parallel training of client-side models while incorporating server feedback via auxiliary models. FSL-SAGE has a finite-time convergence rate of $\mathcal{O}(1/\sqrt{T})$ for $T$ communication rounds, which matches FedAvg. We conducted extensive experiments with large-size computer vision and natural language models to verify the efficacy and the significant amount of communication cost savings of our proposed FSL-SAGE method.

## Acknowledgements

This work is supported in part by ONR grant N00014-24-1-2729; NSF grants CAREER 2110259, 2112471, and 2324052; DARPA YFA D24AP00265 and DARPA HR0011-25-2-0019.

## Impact Statement

This paper presents work whose goal is to advance the field of Machine Learning. There are many potential societal consequences of our work, none of which we feel must be specifically highlighted here.

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

# A. Notation

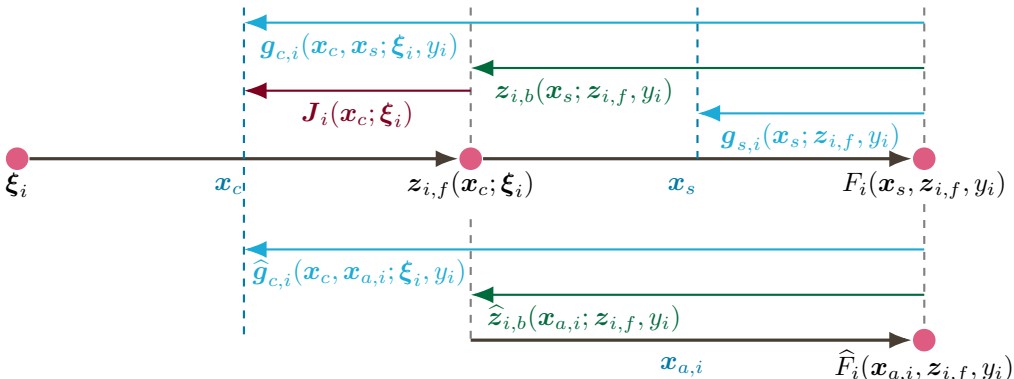

*Figure 6.* Outline and notation of gradient flows in FSL: Blue arrows denote gradients w.r.t. model parameters; green arrows represent gradients w.r.t. the cut-layer; and brown arrow indicates the Jacobian of the cut-layer w.r.t. the client-side parameters.

We use the following additional notation in our convergence analysis. We denote the gradients of the loss $F_i(\boldsymbol{x}; \boldsymbol{\xi}_i, y_i)$ associated with client $i$, w.r.t. the server-side model as $\boldsymbol{g}_{s,i} := \partial F_i / \partial \boldsymbol{x}_s$ and w.r.t. the client-side model as:

$$\boldsymbol{g}_{c,i} := \frac{\partial F_i}{\partial \boldsymbol{x}_c} = \frac{\partial \boldsymbol{z}'_{i,f}}{\partial \boldsymbol{x}_c} \cdot \frac{\partial F_i}{\partial \boldsymbol{z}_{i,f}} = \boldsymbol{J}_{c,i} \cdot \boldsymbol{z}_{i,b}. \tag{A.1}$$

where we use $\boldsymbol{J}_{c,i}$ to denote the transpose of the Jacobian matrix $\partial \boldsymbol{z}_{i,f} / \partial \boldsymbol{x}'_c$ of the cut-layer activations $\boldsymbol{z}_{i,f}$ with respect to the client-side model weights $\boldsymbol{x}_{c,i}$ for the $i^{\text{th}}$ client. The returned gradient $\boldsymbol{z}_{i,b}$ is a function of the model $\boldsymbol{x}$, and the client's mini-batch $(\boldsymbol{\xi}_i, y_i)$, and can be written as $\boldsymbol{z}_{i,b}(\boldsymbol{x}; \boldsymbol{\xi}_i, y_i)$.

We denote the gradients of the loss $F_i(\boldsymbol{x}; \boldsymbol{\xi}_i, y_i)$ associated with client $i$ as $\boldsymbol{g}_i := (\boldsymbol{g}_{c,i}, \boldsymbol{g}_s)$, where $\boldsymbol{g}_{c,i}$ and $\boldsymbol{g}_s$ denote the gradients with respect to the client-side and server-side models respectively.

Figure 6 outlines the gradients used in our analysis. The following equations define the notation used there.

$$\boldsymbol{g}_{s,i}(\boldsymbol{x}_s; \boldsymbol{z}_{i,f}, y_i) := \frac{\partial F_i}{\partial \boldsymbol{x}_s}(\boldsymbol{x}_s; \boldsymbol{z}_{i,f}, y_i) \tag{A.2a}$$

$$\boldsymbol{z}_{i,b}(\boldsymbol{x}_s; \boldsymbol{z}_{i,f}, y_i) := \frac{\partial F_i}{\partial \boldsymbol{z}_{i,f}}(\boldsymbol{x}_s; \boldsymbol{z}_{i,f}, y_i) \tag{A.2b}$$

$$\boldsymbol{J}_i(\boldsymbol{x}_c; \boldsymbol{\xi}_i) := \frac{\partial \boldsymbol{z}'_{i,f}}{\partial \boldsymbol{x}_c}(\boldsymbol{x}_c; \boldsymbol{\xi}_i) \tag{A.2c}$$

$$\boldsymbol{g}_{c,i}(\boldsymbol{x}_c, \boldsymbol{x}_s; \boldsymbol{\xi}_i, y_i) := \boldsymbol{J}_i(\boldsymbol{x}_c; \boldsymbol{\xi}_i) \cdot \boldsymbol{z}_{i,b}(\boldsymbol{x}_s; \boldsymbol{z}_{i,f}, y_i) \tag{A.2d}$$

$$\widehat{\boldsymbol{z}}_{i,b}(\boldsymbol{x}_{a,i}, \boldsymbol{z}_{i,f}, y_i) := \frac{\partial \widehat{F}_i}{\partial \boldsymbol{z}_{i,f}}(\boldsymbol{x}_{a,i}; \boldsymbol{z}_{i,f}, y_i) \tag{A.2e}$$

$$\widehat{\boldsymbol{g}}_{c,i}(\boldsymbol{x}_c, \boldsymbol{x}_{a,i}; \boldsymbol{\xi}_i, y_i) := \boldsymbol{J}_i(\boldsymbol{x}_c; \boldsymbol{\xi}_i) \cdot \widehat{\boldsymbol{z}}_{i,b}(\boldsymbol{x}_{a,i}; \boldsymbol{z}_{i,f}, y_i) \tag{A.2f}$$

Also, we denote the gradient w.r.t. the client-side model with $\nabla_c$ and the w.r.t. the server-side model as $\nabla_s$.

Table 2 summarizes all the notation used in this paper.

# B. Proofs

### B.1. Proof of Theorem 4.3

We first state the following useful lemma that bounds the $k^{\text{th}}$ iterate averaged across clients, which we call the *client drift*. The proof is provided in Appendix B.3.

Table 2. Notations for FSL-SAGE

| Quantity | Meaning |
|---|---|
| $m$ | Number of clients |
| $T$ | Total number of communication rounds |
| $K$ | Number of client local updates |
| $l$ | Frequency of server-side or auxiliary model updates |
| $p := \lfloor \frac{T}{l} \rfloor / T$ | Fraction of rounds for which server-side model is updated |
| $\boldsymbol{x}_c, \boldsymbol{x}_s, \boldsymbol{x}_{a,i}$ | Client-side, server-side and auxiliary model parameters |
| $\widetilde{\boldsymbol{x}}$ | Concatenation of client-side and auxiliary model parameters |
| $\boldsymbol{x}$ | Concatenation of client-side and server-side model parameters |
| $\boldsymbol{g}_{c,i}, \boldsymbol{g}_s, \boldsymbol{g}_i$ | Gradients of $f(\cdot)$ w.r.t. client-, server-side and concatenated model parameters |
| $\eta_L, \eta$ | Learning rates for client- and server-side models respectively |
| $f_0, f^*$ | cost function values at initialization and end of $T$ rounds |
| $\nabla_c, \nabla_s$ | Gradients w.r.t. client- and server-side models |
| $L_c, \widehat{L}_c$ | Lipschitz constants of $\nabla F_i$ and $\nabla \widehat{F}_i$ |
| $L_{zf}$ | Lipschitz constant of $\boldsymbol{z}_{i,f}$ w.r.t. client-side model |
| $\sigma_L$ | Bound on variance of local stochastic gradients |
| $\sigma_G$ | Bound on variance of estimating global cost function gradient |

**Lemma B.1.** *(Client drift bound) For any step-size satisfying $\eta_L \leq \frac{1}{\sqrt{10}K\widehat{L}_c}$, the client drift for any $k \in 0, \ldots, K-1$ can be bounded as:*

$$\frac{1}{m} \sum_{i=1}^{m} \mathbb{E}_t \left[ \left\| \boldsymbol{x}_{c,i}^{t,k} - \boldsymbol{x}_c^t \right\|^2 \right] \leq 20K\eta_L^2 \left[ (\widehat{\sigma}_L^2 + K\sigma_G^2) + K \left\| \nabla_c f(\boldsymbol{x}^t) \right\|^2 \right.$$

$$\left. + \frac{K}{m} \sum_{i=1}^{m} \left\| \nabla_c \widehat{F}_i(\widetilde{\boldsymbol{x}}_i^t) - \nabla_c F_i(\boldsymbol{x}^t) \right\|^2 \right] \tag{B.1}$$

For the following, we denote the expectation conditioned on all randomness up to the $t^{\text{th}}$ step as $\mathbb{E}_t [\cdot]$. First, from Assumption 4.1(a), and the fact that $\boldsymbol{x} := (\boldsymbol{x}_c, \boldsymbol{x}_s)$, we get

$$\mathbb{E}_t \left[ f(\boldsymbol{x}^{t+1}) \right] \leq f(\boldsymbol{x}^t) + \underbrace{\left\langle \nabla_c f(\boldsymbol{x}^t), \mathbb{E}_t \left[ \boldsymbol{x}_c^{t+1} - \boldsymbol{x}_c^t \right] \right\rangle + \frac{L_c}{2} \mathbb{E}_t \left[ \left\| \boldsymbol{x}_c^{t+1} - \boldsymbol{x}_c^t \right\|^2 \right]}_{\triangleq \, \mathcal{C}}$$

$$+ \underbrace{\left\langle \nabla_s f(\boldsymbol{x}^t), \mathbb{E}_t \left[ \boldsymbol{x}_s^{t+1} - \boldsymbol{x}_s^t \right] \right\rangle + \frac{L_c}{2} \mathbb{E}_t \left[ \left\| \boldsymbol{x}_s^{t+1} - \boldsymbol{x}_s^t \right\|^2 \right]}_{\triangleq \, \mathcal{S}} \tag{B.2}$$

Hereafter, the proof is divided into two parts: the client-side bound ($\mathcal{C}$) and the server-side bound ($\mathcal{S}$).

1. **Client-side bound**: At the $k^{\text{th}}$ iteration of the $t^{\text{th}}$ round, the net update for client $i$ takes the form:

$$\boldsymbol{x}_{c,i}^{t,k} = \boldsymbol{x}_{c,i}^{t,k-1} - \eta_L \widehat{\boldsymbol{g}}_{c,i}^{t,k-1} = \boldsymbol{x}_{c,i}^{t,0} - \eta_L \sum_{j=0}^{k-1} \widehat{\boldsymbol{g}}_{c,i}^{t,j}. \tag{B.3}$$

Together with the model averaging at server $F$, the client-side model update in one communication round is given by

$$\boldsymbol{x}_c^{t+1} = \frac{1}{m} \sum_{i=1}^{m} \boldsymbol{x}_{c,i}^{t,K} = \frac{1}{m} \sum_{i=1}^{m} \left[ \boldsymbol{x}_{c,i}^{t,0} - \eta_L \sum_{k=0}^{K-1} \widehat{\boldsymbol{g}}_{c,i}^{t,k} \right] = \boldsymbol{x}_c^t - \frac{\eta_L}{m} \sum_{i=1}^{m} \sum_{k=0}^{K-1} \widehat{\boldsymbol{g}}_{c,i}^{t,k}. \tag{B.4}$$

We first bound the quantity $\mathcal{C}$ in (B.2):

$$\mathcal{C} = \underbrace{\left\langle \nabla_c f(\boldsymbol{x}^t), -\frac{\eta_L}{m}\mathbb{E}_t\left[\sum_{i=1}^{m}\sum_{k=0}^{K-1}\widehat{\boldsymbol{g}}_{c,i}^{t,k}\right]\right\rangle}_{\triangleq A} + \underbrace{\frac{\eta_L^2 L_c}{2}\mathbb{E}_t\left[\left\|\frac{1}{m}\sum_{i=1}^{m}\sum_{k=0}^{K-1}\widehat{\boldsymbol{g}}_{c,i}^{t,k}\right\|^2\right]}_{\triangleq B} \tag{B.5}$$

where we get (B.5) from (B.4). For the sequel, we define $\widetilde{\boldsymbol{x}}_i := (\boldsymbol{x}_{c,i}, \boldsymbol{x}_{a,i})$ as the set of parameters at client $i$. Then, we can bound the expression $A$ in (B.5) as follows.

$$A = \left\langle \nabla_c f(\boldsymbol{x}^t), -\frac{\eta_L}{m}\mathbb{E}_t\left[Km\nabla_c f(\boldsymbol{x}^t) + \sum_{i=1}^{m}\sum_{k=0}^{K-1}\nabla_c\widehat{F}_i(\widetilde{\boldsymbol{x}}_i^{t,k}) - Km\nabla_c f(\boldsymbol{x}^t)\right]\right\rangle$$

$$= -K\eta_L\left\|\nabla_c f(\boldsymbol{x}^t)\right\|^2 + \left\langle \nabla_c f(\boldsymbol{x}^t), -\frac{\eta_L}{m}\sum_{i=1}^{m}\sum_{k=0}^{K-1}\mathbb{E}_t\left[\nabla_c\widehat{F}_i(\widetilde{\boldsymbol{x}}_i^{t,k}) - \nabla_c F_i(\boldsymbol{x}^t)\right]\right\rangle$$

$$= -K\eta_L\left\|\nabla_c f(\boldsymbol{x}^t)\right\|^2$$
$$+ \mathbb{E}_t\left[\left\langle \sqrt{K\eta_L}\nabla_c f(\boldsymbol{x}^t), -\frac{\sqrt{\eta_L}}{m\sqrt{K}}\sum_{i=1}^{m}\sum_{k=0}^{K-1}\left\{\nabla_c\widehat{F}_i(\widetilde{\boldsymbol{x}}_i^{t,k}) - \nabla_c F_i(\boldsymbol{x}^t)\right\}\right\rangle\right] \tag{B.6}$$

Using the property $\langle a, b\rangle = 1/2\left\|a\right\|^2 + 1/2\left\|b\right\|^2 - 1/2\left\|a - b\right\|^2$, we can rewrite $A$ as

$$A = -\frac{K\eta_L}{2}\left\|\nabla_c f(\boldsymbol{x}^t)\right\|^2 + \frac{\eta_L}{2m^2 K}\mathbb{E}_t\left[\left\|\sum_{i=1}^{m}\sum_{k=0}^{K-1}\left\{\nabla_c\widehat{F}_i(\widetilde{\boldsymbol{x}}_i^{t,k}) - \nabla_c F_i(\boldsymbol{x}^t)\right\}\right\|^2\right]$$

$$- \frac{\eta_L}{2m^2 K}\mathbb{E}_t\left[\left\|mK\nabla_c f(\boldsymbol{x}^t) + \sum_{i=1}^{m}\sum_{k=0}^{K-1}\left\{\nabla_c\widehat{F}_i(\widetilde{\boldsymbol{x}}_i^{t,k}) - \nabla_c F_i(\boldsymbol{x}^t)\right\}\right\|^2\right]$$

$$= -\frac{K\eta_L}{2}\left\|\nabla_c f(\boldsymbol{x}^t)\right\|^2 + \frac{\eta_L}{2m^2 K}\mathbb{E}_t\left[\left\|\sum_{i=1}^{m}\sum_{k=0}^{K-1}\left\{\nabla_c\widehat{F}_i(\widetilde{\boldsymbol{x}}_i^{t,k}) - \nabla_c F_i(\boldsymbol{x}^t)\right\}\right\|^2\right]$$

$$- \frac{\eta_L}{2m^2 K}\mathbb{E}_t\left[\left\|\sum_{i=1}^{m}\sum_{k=0}^{K-1}\nabla_c\widehat{F}_i(\widetilde{\boldsymbol{x}}_i^{t,k})\right\|^2\right]$$

$$\leq -\frac{K\eta_L}{2}\left\|\nabla_c f(\boldsymbol{x}^t)\right\|^2 + \frac{\eta_L}{2m}\sum_{i=1}^{m}\sum_{k=0}^{K-1}\underbrace{\mathbb{E}_t\left[\left\|\nabla_c\widehat{F}_i(\widetilde{\boldsymbol{x}}_i^{t,k}) - \nabla_c F_i(\boldsymbol{x}^t)\right\|^2\right]}_{\triangleq A_1}$$

$$- \frac{\eta_L}{2m^2 K}\mathbb{E}_t\left[\left\|\sum_{i=1}^{m}\sum_{k=0}^{K-1}\nabla_c\widehat{F}_i(\widetilde{\boldsymbol{x}}_i^{t,k})\right\|^2\right] \tag{B.7}$$

Next, we bound $A_1$ in (B.7) as follows:

$$A_1 = \mathbb{E}_t\left[\left\|\nabla_c\widehat{F}_i(\boldsymbol{x}_i^{t,k}) - \nabla_c F_i(\boldsymbol{x}^t)\right\|^2\right]$$

$$= \mathbb{E}_t\left[\left\|\nabla_c\widehat{F}_i(\widetilde{\boldsymbol{x}}_i^{t,k}) - \nabla_c\widehat{F}_i(\widetilde{\boldsymbol{x}}_i^{t,0}) + \nabla_c\widehat{F}_i(\widetilde{\boldsymbol{x}}_i^{t,0}) - \nabla_c F_i(\boldsymbol{x}^t)\right\|^2\right]$$

$$\leq 2\mathbb{E}_t\left[\left\|\nabla_c\widehat{F}_i(\widetilde{\boldsymbol{x}}_i^{t,k}) - \nabla_c\widehat{F}_i(\widetilde{\boldsymbol{x}}_i^{t,0})\right\|^2\right] + 2\mathbb{E}_t\left[\left\|\nabla_c\widehat{F}_i(\widetilde{\boldsymbol{x}}_i^{t,0}) - \nabla_c F_i(\boldsymbol{x}^t)\right\|^2\right] \tag{B.8}$$

$$\leq 2\widehat{L}_c^2\mathbb{E}_t\left[\left\|\boldsymbol{x}_{c,i}^{t,k} - \boldsymbol{x}_c^t\right\|^2\right] + 2\left\|\nabla_c\widehat{F}_i(\widetilde{\boldsymbol{x}}_i^t) - \nabla_c F_i(\boldsymbol{x}^t)\right\|^2 \tag{B.9}$$

where, $n := \lfloor t/l \rfloor$ in (B.8) we use the property $\left\| \sum_{i=1}^{n} \boldsymbol{a}_i \right\|^2 \leq n \sum_{i=1}^{n} \|\boldsymbol{a}_i\|^2$ with $n = 2$, in (B.9) we use Assumption 4.1(b). Next, bounding the quantity $B$ in (B.5):

$$
\begin{aligned}
B &= \frac{\eta_L^2 L_c}{2} \mathbb{E}_t \left[ \left\| \frac{1}{m} \sum_{i=1}^{m} \sum_{k=0}^{K-1} \widehat{\boldsymbol{g}}_{c,i}^{t,k} \right\|^2 \right] \\
&= \frac{\eta_L^2 L_c}{2} \mathbb{E}_t \left[ \left\| \frac{1}{m} \sum_{i=1}^{m} \sum_{k=0}^{K-1} \left\{ \widehat{\boldsymbol{g}}_{c,i}^{t,k} - \nabla_c \widehat{F}_i(\widetilde{\boldsymbol{x}}_i^{t,k}) \right\} \right\|^2 \right] + \frac{\eta_L^2 L_c}{2m^2} \mathbb{E}_t \left[ \left\| \sum_{i=1}^{m} \sum_{k=0}^{K-1} \nabla_c \widehat{F}_i(\widetilde{\boldsymbol{x}}_i^{t,k}) \right\|^2 \right] &\text{(B.10)} \\
&\leq \frac{\eta_L^2 K \widehat{\sigma}_L^2 L_c}{2m} + \frac{\eta_L^2 L_c}{2m^2} \mathbb{E}_t \left[ \left\| \sum_{i=1}^{m} \sum_{k=0}^{K-1} \nabla_c \widehat{F}_i(\widetilde{\boldsymbol{x}}_i^{t,k}) \right\|^2 \right] &\text{(B.11)}
\end{aligned}
$$

where (B.10) follows from $\mathbb{E}\left[ \|x\|^2 \right] = \mathbb{E}\left[ \|x - \mathbb{E}[x]\|^2 \right] + \|\mathbb{E}[x]\|^2$, and (B.11) from Assumption 4.2(a). Substituting (B.9) in (B.7), and then (B.7) and (B.11) in (B.5) we get the following.

$$
\begin{aligned}
\mathcal{C} &= A + B \\
&\leq -\frac{K\eta_L}{2} \|\nabla_c f(\boldsymbol{x}^t)\|^2 + \frac{\eta_L^2 K \widehat{\sigma}_L^2 L_c}{2m} \\
&\quad + \frac{\eta_L}{m} \sum_{i=1}^{m} \sum_{k=0}^{K-1} \left[ \widehat{L}_c^2 \mathbb{E}\left[ \|\boldsymbol{x}_{c,i}^{t,k} - \boldsymbol{x}_c^t\|^2 \right] + \|\nabla_c \widehat{F}_i(\widetilde{\boldsymbol{x}}_i^t) - \nabla_c F_i(\boldsymbol{x}^t)\|^2 \right] \\
&\quad - \frac{\eta_L}{2m^2 K}(1 - \eta_L L_c K) \mathbb{E}_t \left[ \left\| \sum_{i=1}^{m} \sum_{k=0}^{K-1} \nabla_c \widehat{F}_i(\widetilde{\boldsymbol{x}}_i^{t,k}) \right\|^2 \right] &\text{(B.12)} \\
&\leq -\frac{K\eta_L}{2} \|\nabla_c f(\boldsymbol{x}^t)\|^2 + \frac{\eta_L^2 K \widehat{\sigma}_L^2 L_c}{2m} + \eta_L \widehat{L}_c^2 \sum_{k=0}^{K-1} \left\{ \frac{1}{m} \sum_{i=1}^{m} \mathbb{E}\left[ \|\boldsymbol{x}_{c,i}^{t,k} - \boldsymbol{x}_c^t\|^2 \right] \right\} \\
&\quad + \frac{\eta_L K}{m} \sum_{i=1}^{m} \|\nabla_c \widehat{F}_i(\widetilde{\boldsymbol{x}}_i^t) - \nabla_c F_i(\boldsymbol{x}^t)\|^2 &\text{(B.13)}
\end{aligned}
$$

where we use the fact that the last term in (B.12) is strictly negative for $\eta_L < \frac{1}{KL_c}$ to get (B.13). In (B.13), using Lemma B.1 and rearranging, we get:

$$
\begin{aligned}
\mathcal{C} &\leq -\frac{K\eta_L}{2} \|\nabla_c f(\boldsymbol{x}^t)\|^2 + \frac{\eta_L^2 K \widehat{\sigma}_L^2 L_c}{2m} + \frac{\eta_L K}{m} \sum_{i=1}^{m} \|\nabla_c \widehat{F}_i(\widetilde{\boldsymbol{x}}_i^t) - \nabla_c F_i(\boldsymbol{x}^t)\|^2 \\
&\quad + 20K^2 \eta_L^3 \widehat{L}_c^2 \left[ (\widehat{\sigma}_L^2 + K\sigma_G^2) + K \|\nabla_c f(\boldsymbol{x}^t)\|^2 + \frac{K}{m} \sum_{i=1}^{m} \|\nabla_c \widehat{F}_i(\widetilde{\boldsymbol{x}}_i^t) - \nabla_c F_i(\boldsymbol{x}^t)\|^2 \right] \\
&= -K\eta_L \left( \frac{1}{2} - 20K^2 \eta_L^2 \widehat{L}_c^2 \right) \|\nabla_c f(\boldsymbol{x}^t)\|^2 + \frac{\eta_L^2 K \widehat{\sigma}_L^2 L_c}{2m} + 20K^2 \eta_L^3 \widehat{L}_c^2 (\widehat{\sigma}_L^2 + K\sigma_G^2) \\
&\quad + \frac{K\eta_L}{m}(1 + 20K^2 \eta_L^2 \widehat{L}_c^2) \sum_{i=1}^{m} \|\nabla_c \widehat{F}_i(\widetilde{\boldsymbol{x}}_i^t) - \nabla_c F_i(\boldsymbol{x}^t)\|^2 \\
&\leq -cK\eta_L \|\nabla_c f(\boldsymbol{x}^t)\|^2 + \frac{\eta_L^2 K \widehat{\sigma}_L^2 L_c}{2m} + 20K^2 \eta_L^3 \widehat{L}_c^2 (\widehat{\sigma}_L^2 + K\sigma_G^2) \\
&\quad + \frac{3K\eta_L}{2m} \sum_{i=1}^{m} \|\nabla_c \widehat{F}_i(\widetilde{\boldsymbol{x}}_i^t) - \nabla_c F_i(\boldsymbol{x}^t)\|^2 &\text{(B.14)}
\end{aligned}
$$

We get (B.14) because, $\exists\, c > 0$ such that $(0.5 - 20K^2 \eta_L^2 \widehat{L}_c^2) > c$ provided $\eta_L < \frac{1}{2\sqrt{10}K\widehat{L}_c}$, and by reusing the bound on $\eta_L$ in the last term as well. Rewriting (B.14) in terms of $\Phi_1(\eta_L) := \frac{1}{c} \left[ \frac{\eta_L \widehat{\sigma}_L^2 L_c}{2m} + 20K\eta_L^2 \widehat{L}_c^2 (\widehat{\sigma}_L^2 + K\sigma_G^2) \right]$ and the

estimation error metric $\varepsilon^t$ in (5), we get:

$$\mathcal{C} \le cK\eta_L \left( -\left\| \nabla_c f(\boldsymbol{x}^t) \right\|^2 + \frac{3}{2c}\varepsilon^t + \Phi_1(\eta_L) \right). \tag{B.15}$$

2. **Server-side bound**: The clients transmit the smashed data to the $S$-server once every $K/Q$ local iterations, thus the server-side model is updated $Q$ times per round per client. The update equation for the server-side model is then:

$$\boldsymbol{x}_s^{t+1} = \boldsymbol{x}_s^t - \eta \sum_{i=1}^m \sum_{q=0}^{Q-1} \boldsymbol{g}_{s,i}^{t,q} \tag{B.16}$$

where the second superscript on $\boldsymbol{g}_{s,i}^{t,q}$ indicates the $q^{th}$ server update, and is different from the local client iteration index $k$. Then, the server-side bound $\mathcal{S}$ over one communication round in (B.2) can be written as follows:

$$\begin{aligned}
\mathcal{S} &= \left\langle \nabla_s f(\boldsymbol{x}^t), \mathbb{E}_t \left[ \boldsymbol{x}_s^{t+1} - \boldsymbol{x}_s^t \right] \right\rangle + \frac{L_c}{2} \mathbb{E}_t \left[ \left\| \boldsymbol{x}_s^{t+1} - \boldsymbol{x}_s^t \right\|^2 \right] \\
&= \mathbb{E}_t \left[ \left\langle \nabla_s f(\boldsymbol{x}^t), -\eta \sum_{i=1}^m \sum_{q=0}^{Q-1} \boldsymbol{g}_{s,i}^{t,q} \right\rangle \right] + \frac{\eta^2 L_c}{2} \mathbb{E}_t \left[ \left\| \sum_{i=1}^m \sum_{q=0}^{Q-1} \boldsymbol{g}_{s,i}^{t,q} \right\|^2 \right]
\end{aligned} \tag{B.17}$$

Equation (B.17) is identical to (B.5) with the substitutions $\eta_L/m \to \eta$, $\widehat{\boldsymbol{g}}_{c,i}^{t,k} \to \boldsymbol{g}_{s,i}^{t,q}$, $\nabla_c \to \nabla_s$ and $K \to Q$. Following the same steps as in the client-side bound we can say that under Assumptions 4.1(a) and 4.2(b), and provided $\eta \le \frac{1}{4\sqrt{2}QL_c}$, $\mathcal{S}$ can be bounded as

$$\mathcal{S} \le c'mQ\eta \left( -\left\| \nabla_s f(\boldsymbol{x}^t) \right\|^2 + \Phi_2(\eta) \right) \tag{B.18}$$

with $\Phi_2(\eta) := \frac{1}{c'} \left[ \frac{\eta\sigma_L^2 L_c}{2} + 16Q\eta^2 L_c^2(\sigma_L^2 + Q\sigma_G^2) \right]$.

Finally, we can substitute (B.15) and (B.18) back in (B.2) to get the following:

$$\mathbb{E}_t \left[ f(\boldsymbol{x}^{t+1}) \right] \le f(\boldsymbol{x}^t) + cK\eta_L \left( -\left\| \nabla_c f(\boldsymbol{x}^t) \right\|^2 + \frac{3}{2c}\varepsilon^t + \Phi_1(\eta_L) \right) + c'mQ\eta \left( -\left\| \nabla_s f(\boldsymbol{x}^t) \right\|^2 + \Phi_2(\eta) \right)$$

which can be rearranged to get:

$$cK\eta_L \left\| \nabla_c f(\boldsymbol{x}^t) \right\|^2 + c'mQ\eta \left\| \nabla_s f(\boldsymbol{x})^t \right\|^2 \le f(\boldsymbol{x}^t) - \mathbb{E}_t \left[ f(\boldsymbol{x}^{t+1}) \right] + \frac{3K\eta_L}{2}\varepsilon^t + cK\eta_L\Phi_1(\eta_L) + c'Qm\eta\Phi_2(\eta)$$

Next, by redefining $c = \min\{c, c'\}$ and $c' = \max\{c, c'\}$. After further rearranging we get:

$$cQ\min\{\eta_L, m\eta\} \left\| \nabla f(\boldsymbol{x}^t) \right\|^2 \le f(\boldsymbol{x}^t) - \mathbb{E}_t \left[ f(\boldsymbol{x}^{t+1}) \right] + \frac{3K\eta_L}{2}\varepsilon^t + c'K \left[ \eta_L\Phi_1(\eta_L) + m\eta\Phi_2(\eta) \right]$$

$$\implies \left\| \nabla f(\boldsymbol{x}^t) \right\|^2 \le \frac{f(\boldsymbol{x}^t) - \mathbb{E}_t \left[ f(\boldsymbol{x}^{t+1}) \right]}{cQ\min\{\eta_L, m\eta\}} + \frac{3K\eta_L}{2cQ\min\{\eta_L, m\eta\}}\varepsilon^t + \frac{c'K}{cQ} \left( \frac{\eta_L\Phi_1(\eta_L) + m\eta\Phi_2(\eta)}{\min\{\eta_L, m\eta\}} \right)$$

Next, taking full expectation on both sides, and summing over $t = 1, 2, \dots, T$ and recognizing that minimum is lesser than the average, we get the final bound:

$$\min_{t\in[T]} \mathbb{E} \left[ \left\| \nabla f(\boldsymbol{x}^t) \right\|^2 \right] \le \frac{f(\boldsymbol{x}^0) - f^*}{c\min\{\eta_L, m\eta\}QT} + \frac{3K\eta_L}{2cQ\min\{\eta_L, m\eta\}} \frac{1}{T} \sum_{t=1}^T \varepsilon^t + \frac{\widetilde{\Phi}(\eta_L, \eta)}{T} \tag{B.19}$$

where $\widetilde{\Phi}(\eta_L, \eta)$ is given by:

$$\begin{aligned}
\widetilde{\Phi}(\eta_L, \eta) &:= \frac{c'K}{cQ} \cdot \frac{\frac{\eta_L}{c} \left[ \frac{\eta_L\widehat{\sigma}_L^2 L_c}{2m} + 20K\eta_L^2\widehat{L}_c^2(\widehat{\sigma}_L^2 + K\sigma_G^2) \right] + \frac{m\eta}{c'} \left[ \frac{\eta\sigma_L^2 L_c}{2} + 16Q\eta^2 L_c^2(\sigma_L^2 + Q\sigma_G^2) \right]}{\min\{\eta_L, m\eta\}} \\
&\le c'K \cdot \frac{\frac{\eta_L^2\widehat{\sigma}_L^2 L_c}{2m} + 20K\eta_L^3\widehat{L}_c^2(\widehat{\sigma}_L^2 + K\sigma_G^2) + \frac{m\eta^2\sigma_L^2 L_c}{2} + 16Qm\eta^3 L_c^2(\sigma_L^2 + Q\sigma_G^2)}{c^2Q\min\{\eta_L, m\eta\}} =: \Phi(\eta_L, \eta) \tag{B.20}
\end{aligned}$$

Thus, substituting $\Phi(\eta_L, \eta)$ instead of $\widetilde{\Phi}(\eta_L, \eta)$ in (B.19) we get the statement of the theorem. $\square$

## B.2. Proof of Theorem 4.8 and Corollary 4.10

The statement of the theorem follows by further bounding the estimator term $\frac{1}{T}\sum_{t=0}^{T-1}\varepsilon^t$ in the RHS in Theorem 4.3. There, we obtained an upper bound on the stationarity gap, i.e., $\min_t \|\nabla f(\boldsymbol{x}_t)\|^2$, which contained a term involving

$$\varepsilon^t := \frac{1}{m}\sum_{i=1}^{m}\mathbb{E}\left[\left\|\nabla_c\widehat{F}_i(\widetilde{\boldsymbol{x}}_i^t) - \nabla_c F_i(\boldsymbol{x}^t)\right\|^2\right] \tag{B.21}$$

where the gradients $\nabla_c\widehat{F}_i(\widetilde{\boldsymbol{x}}_i^t)$ and $\nabla_c F_i(\boldsymbol{x}_i^t)$ are written in terms of the cut-layer activations as:

$$\nabla_c\widehat{F}_i(\widetilde{\boldsymbol{x}}^t) := \mathbb{E}_t\left[\boldsymbol{J}_i(\boldsymbol{x}_c^t;\boldsymbol{\xi}_i^t)\cdot\widehat{\boldsymbol{z}}_{i,b}(\boldsymbol{x}_{a,i}^t;\boldsymbol{z}_{i,f}^t,y_i^t)\right] \tag{B.22}$$

$$\nabla_c F_i(\boldsymbol{x}^t) := \mathbb{E}_t\left[\boldsymbol{J}_i(\boldsymbol{x}_c^t;\boldsymbol{\xi}_i)\cdot\boldsymbol{z}_{i,b}(\boldsymbol{x}_s^t;\boldsymbol{z}_{i,f}^t,y_i^t)\right] \tag{B.23}$$

where $\mathbb{E}_t[\cdot]$ refers to expectation with respect to the randomness at iteration $t$. We can then bound $\varepsilon^t$ as follows:

$$\varepsilon_t = \frac{1}{m}\sum_{i=1}^{m}\mathbb{E}\left[\left\|\nabla_c\widehat{F}_i(\widetilde{\boldsymbol{x}}_i^t) - \nabla_c F_i(\boldsymbol{x}^t)\right\|^2\right]$$

$$= \frac{1}{m}\sum_{i=1}^{m}\mathbb{E}\left[\left\|\mathbb{E}_t\left[\boldsymbol{J}_i(\boldsymbol{x}_c^t;\boldsymbol{\xi}_i^t)\cdot\left(\widehat{\boldsymbol{z}}_{i,b}(\boldsymbol{x}_{a,i}^t;\boldsymbol{z}_{i,f}^t,y_i^t) - \boldsymbol{z}_{i,b}(\boldsymbol{x}_s^t;\boldsymbol{z}_{i,f}^t,y_i^t))\right]\right\|^2\right]$$

$$\leq \frac{1}{m}\sum_{i=1}^{m}\mathbb{E}\left[\left\|\boldsymbol{J}_i(\boldsymbol{x}_c^t;\boldsymbol{\xi}_i^t)\cdot\left(\widehat{\boldsymbol{z}}_{i,b}(\boldsymbol{x}_{a,i}^t;\boldsymbol{z}_{i,f}^t,y_i^t) - \boldsymbol{z}_{i,b}(\boldsymbol{x}_s^t;\boldsymbol{z}_{i,f}^t,y_i^t))\right\|^2\right] \tag{B.24}$$

$$\leq \frac{1}{m}\sum_{i=1}^{m}\mathbb{E}\left[\left\|\boldsymbol{J}_i(\boldsymbol{x}_c^t;\boldsymbol{\xi}_i^t)\right\|^2\cdot\left\|\widehat{\boldsymbol{z}}_{i,b}(\boldsymbol{x}_{a,i}^t;\boldsymbol{z}_{i,f}^t,y_i^t) - \boldsymbol{z}_{i,b}(\boldsymbol{x}_s^t;\boldsymbol{z}_{i,f}^t,y_i^t)\right\|^2\right] \tag{B.25}$$

$$\leq \frac{L_f^2}{m}\sum_{i=1}^{m}\mathbb{E}\left[\left\|\widehat{\boldsymbol{z}}_{i,b}(\boldsymbol{x}_{a,i}^t;\boldsymbol{z}_{i,f}^t,y_i^t) - \boldsymbol{z}_{i,b}(\boldsymbol{x}_s^t;\boldsymbol{z}_{i,f}^t,y_i^t)\right\|^2\right] \tag{B.26}$$

$$= \frac{L_f^2}{m}\sum_{i=1}^{m}\mathcal{L}_i(\boldsymbol{x}_{a,i}^t,\boldsymbol{x}^t)$$

where we use Jensen's inequality in (B.24) and the spectral norm of $\boldsymbol{J}_i(\cdot;\cdot)$ in (B.25). In (B.26) we use Assumption 4.7 and the fact that the spectral norm of the Jacobian is bounded by the Lipschitz constant.

Note that $r_i(\epsilon) = \mathcal{O}\left(1/\epsilon^2\right)$ implies that $\epsilon = \mathcal{O}\left(1/\sqrt{\tau}\right) = \mathcal{O}\left(1/\sqrt{t}\right)$. Now, from (13), we have:

$$\varepsilon_t \leq \underbrace{\frac{L_f^2}{m}\sum_{i=1}^{m}\mathcal{L}_i(\boldsymbol{x}_{a,i}^{t\star},\boldsymbol{x}^t)}_{L_f^2\varepsilon_\star^t} + \frac{C_1}{\sqrt{t}} \tag{B.27}$$

for some $C_1 > 0$. Note that $\varepsilon_\star^t$ is the lowest achievable error rate by the hypothesis class $\mathcal{A}_i$ at time $t$ and is not reducible further. Finally, substituting (B.27) back into (6) and using $1/T\sum_{t=1}^{T}1/\sqrt{t} \leq 2/\sqrt{T}$ from Lemma B.2, we get the desired result in Theorem 4.8.

For the proof of Corollary 4.10, the only difference is that we need to bound the round average of $\mathbb{E}[\varepsilon_t]$ as follows:

$$
\begin{aligned}
\frac{1}{T}\sum_{t=0}^{T-1}\mathbb{E}[\varepsilon_t] &\leq \frac{1}{T}\sum_{t=0}^{T-1}\left[L_f^2\varepsilon_\star^t + \frac{C_1}{\sqrt{t}}\right]\\
&= \frac{L_F^2}{T}\sum_{t=0}^{T-1}\varepsilon_\star^t + \frac{C_1}{T}\sum_{t=0}^{T-1}\frac{1}{\sqrt{t}}\\
&= \frac{L_F^2}{T}\sum_{t=0}^{T-1}\varepsilon_\star^t + C_1\left\{\frac{1}{T}\sum_{t=0}^{T'-1}\frac{1}{\sqrt{t}} + \frac{1}{T}\sum_{t=T'}^{T-1}\frac{1}{\sqrt{T'}}\right\}\\
&\leq \frac{L_F^2}{T}\sum_{t=0}^{T-1}\varepsilon_\star^t + C_1\left\{\frac{2T'/T}{\sqrt{T'}} + \frac{(1-T'/T)}{\sqrt{T'}}\right\}\\
&= \frac{L_F^2}{T}\sum_{t=0}^{T-1}\varepsilon_\star^t + C_1\frac{(1+T'/T)}{\sqrt{T'}}
\end{aligned}
$$

for some $C_1 > 0$. Note that when $T' = T$, alignment happens till the last round, and this recovers the bound in Theorem 4.8. On the other hand when $T' = 0$, no alignment occurs, and this makes the upper bound $\to \infty$. $\qquad\square$

## B.3. Proof of Lemma B.1

The proof is along similar lines as (Reddi et al., 2021) and proceeds as follows. We start with the iterate:

$$
\begin{aligned}
\mathbb{E}_t\left[\left\|\boldsymbol{x}_{c,i}^{t,k} - \boldsymbol{x}_c^t\right\|^2\right] &= \mathbb{E}_t\left[\left\|\boldsymbol{x}_{c,i}^{t,k-1} - \boldsymbol{x}_c^t - \eta_L\widehat{\boldsymbol{g}}_{c,i}^{t,k-1}\right\|^2\right]\\
&= \mathbb{E}_t\left[\left\|\boldsymbol{x}_{c,i}^{t,k-1} - \boldsymbol{x}_c^t - \eta_L\left\{\widehat{\boldsymbol{g}}_{c,i}^{t,k-1} - \nabla_c\widehat{F}_i(\widetilde{\boldsymbol{x}}_i^{t,k-1}) + \nabla_c\widehat{F}_i(\widetilde{\boldsymbol{x}}_i^{t,k-1}) - \nabla_c\widehat{F}_i(\widetilde{\boldsymbol{x}}_i^t)\right.\right.\right.\\
&\qquad\left.\left.\left. + \nabla_c\widehat{F}_i(\widetilde{\boldsymbol{x}}_i^t) - \nabla_cF_i(\boldsymbol{x}^t) + \nabla_cF_i(\boldsymbol{x}^t) - \nabla_cf(\boldsymbol{x}^t) + \nabla_cf(\boldsymbol{x}^t)\right\}\right\|^2\right]\\
&\leq \mathbb{E}_t\left[\left\|\boldsymbol{x}_{c,i}^{t,k-1} - \boldsymbol{x}_c^t\right\|^2\right] + 5\eta_L^2\mathbb{E}_t\left[\left\|\widehat{\boldsymbol{g}}_{c,i}^{t,k-1} - \nabla_c\widehat{F}_i(\widetilde{\boldsymbol{x}}_i^{t,k-1})\right\|^2\right]\\
&\quad + 5\eta_L^2\mathbb{E}_t\left[\left\|\nabla_c\widehat{F}_i(\widetilde{\boldsymbol{x}}_i^{t,k-1}) - \nabla_c\widehat{F}_i(\widetilde{\boldsymbol{x}}_i^t)\right\|^2\right] + 5\eta_L^2\left\|\nabla_c\widehat{F}_i(\widetilde{\boldsymbol{x}}_i^t) - \nabla_cF_i(\boldsymbol{x}^t)\right\|^2\\
&\quad + 5\eta_L^2\left\|\nabla_cF_i(\boldsymbol{x}^t) - \nabla_cf(\boldsymbol{x}^t)\right\|^2 + 5\eta_L^2\left\|\nabla_cf(\boldsymbol{x}^t)\right\|^2 \qquad\text{(B.28)}
\end{aligned}
$$

where in (B.28), we use the property that $\|\sum_i \boldsymbol{a}_i\|^2 \leq n\sum_i \|\boldsymbol{a}_i^2\|$. Since $K > 1$, we can continue bounding the expression on the right hand side as follows:

$$
\begin{aligned}
\mathbb{E}_t\left[\left\|\boldsymbol{x}_{c,i}^{t,k} - \boldsymbol{x}_c^t\right\|^2\right] &\leq \left(1 + \frac{1}{2K-1}\right)\mathbb{E}_t\left[\left\|\boldsymbol{x}_{c,i}^{t,k-1} - \boldsymbol{x}_c^t\right\|^2\right] + 5\eta_L^2\mathbb{E}_t\left[\left\|\widehat{\boldsymbol{g}}_{c,i}^{t,k-1} - \nabla_c\widehat{F}_i(\widetilde{\boldsymbol{x}}_i^{t,k-1})\right\|^2\right]\\
&\quad + 5K\eta_L^2\mathbb{E}_t\left[\left\|\nabla_c\widehat{F}_i(\widetilde{\boldsymbol{x}}_i^{t,k-1}) - \nabla_c\widehat{F}_i(\widetilde{\boldsymbol{x}}_i^t)\right\|^2\right] + 5K\eta_L^2\left\|\nabla_c\widehat{F}_i(\widetilde{\boldsymbol{x}}_i^t) - \nabla_cF_i(\boldsymbol{x}^t)\right\|^2\\
&\quad + 5K\eta_L^2\left\|\nabla_cF_i(\boldsymbol{x}^t) - \nabla_cf(\boldsymbol{x}^t)\right\|^2 + 5K\eta_L^2\left\|\nabla_cf(\boldsymbol{x}^t)\right\|^2\\
&\leq \left(1 + \frac{1}{2K-1} + 5K\eta_L^2\widehat{L}_c^2\right)\mathbb{E}_t\left[\left\|\boldsymbol{x}_{c,i}^{t,k-1} - \boldsymbol{x}_c^t\right\|^2\right] + 5\eta_L^2(\widehat{\sigma}_L^2 + K\sigma_G^2)\\
&\quad + 5K\eta_L^2\left\|\nabla_c\widehat{F}_i(\widetilde{\boldsymbol{x}}_i^t) - \nabla_cF_i(\boldsymbol{x}^t)\right\|^2 + 5K\eta_L^2\left\|\nabla_cf(\boldsymbol{x}^t)\right\|^2 \qquad\text{(B.29)}\\
&\leq \left(1 + \frac{1}{K-1}\right)\mathbb{E}_t\left[\left\|\boldsymbol{x}_{c,i}^{t,k-1} - \boldsymbol{x}_c^t\right\|^2\right] + 5\eta_L^2(\widehat{\sigma}_L^2 + K\sigma_G^2)\\
&\quad + 5K\eta_L^2\left\|\nabla_c\widehat{F}_i(\widetilde{\boldsymbol{x}}_i^t) - \nabla_cF_i(\boldsymbol{x}^t)\right\|^2 + 5K\eta_L^2\left\|\nabla_cf(\boldsymbol{x}^t)\right\|^2 \qquad\text{(B.30)}
\end{aligned}
$$

where in (B.29), we use Assumptions 4.2 and 4.1(b); and (B.30) holds when $\eta_L \leq \frac{1}{\sqrt{10K\widehat{L}_c}}$. Next, averaging over $m$ clients and unrolling the recursion we can obtain

$$\frac{1}{m} \sum_{i=1}^{m} \mathbb{E}_t \left[ \left\| \boldsymbol{x}_{c,i}^{t,k} - \boldsymbol{x}_c^t \right\|^2 \right] \leq \sum_{p=0}^{k-1} \left( 1 + \frac{1}{K-1} \right)^p \left[ 5\eta_L^2 (\widehat{\sigma}_L^2 + K\sigma_G^2) + 5K\eta_L^2 \left\| \nabla_c f(\boldsymbol{x}^t) \right\|^2 \right.$$
$$\left. + \frac{5K\eta_L^2}{m} \sum_{i=1}^{m} \left\| \nabla_c \widehat{F}_i(\widetilde{\boldsymbol{x}}_i^t) - \nabla_c F_i(\boldsymbol{x}^t) \right\|^2 \right] \tag{B.31}$$

Representing the term in square brackets as $H^t$, we can write:

$$\frac{1}{m} \sum_{i=1}^{m} \mathbb{E}_t \left[ \left\| \boldsymbol{x}_{c,i}^{t,k} - \boldsymbol{x}_c^t \right\|^2 \right] \leq (K-1) \left[ \left( 1 + \frac{1}{K-1} \right)^k - 1 \right] H^t$$

$$\leq (K-1) \left[ \left( 1 + \frac{1}{K-1} \right)^K - 1 \right] H^t \tag{B.32}$$

$$\leq 4KH^t \tag{B.33}$$

where, we use the fact that $k \leq K$ and that $1 + 1/(K-1) \geq 1$ in (B.32), and the fact that for $K > 1$, $\{1 + 1/(K-1)\}^K < 5$ in (B.33). Finally, substituting the value of $H^t$ in (B.33), we obtain the inequality in the lemma. $\square$

## B.4. Additional Lemmas

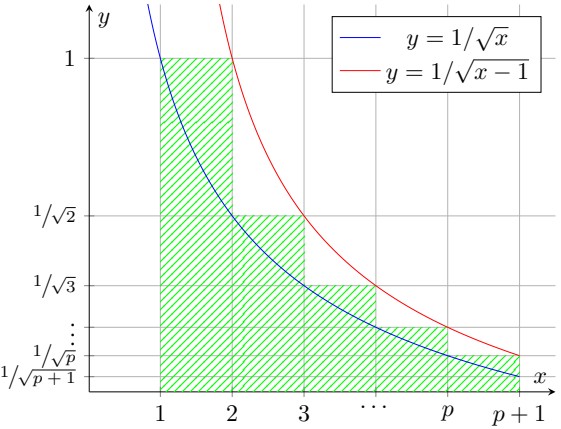

Figure 7. Illustration of the inequality in (B.36)

**Lemma B.2** (Series bound). *Given $p \in \mathbb{N}$, the series:*

$$S(p) \triangleq \frac{1}{p} \sum_{r=1}^{p} \frac{1}{\sqrt{r}} \tag{B.34}$$

*satisfies the bounds*

$$\frac{2}{\sqrt{p+1}+1} \leq S(p) \leq \frac{2}{\sqrt{p}}. \tag{B.35}$$

*In other words $S(p) = \mathcal{O}\left(\frac{1}{\sqrt{p}}\right)$.*

*Proof.* The series in (B.34) can be bounded using integrals as follows:

$$\frac{1}{p} \int_1^{p+1} \frac{dx}{\sqrt{x}} \leq S(p) \leq \frac{1}{p} \int_1^{p+1} \frac{dx}{\sqrt{x-1}} \tag{B.36}$$

The reason for this becomes clear from the illustration in Figure 7. The integrals in (B.36) simplifies to the desired lower and upper bounds. $\square$

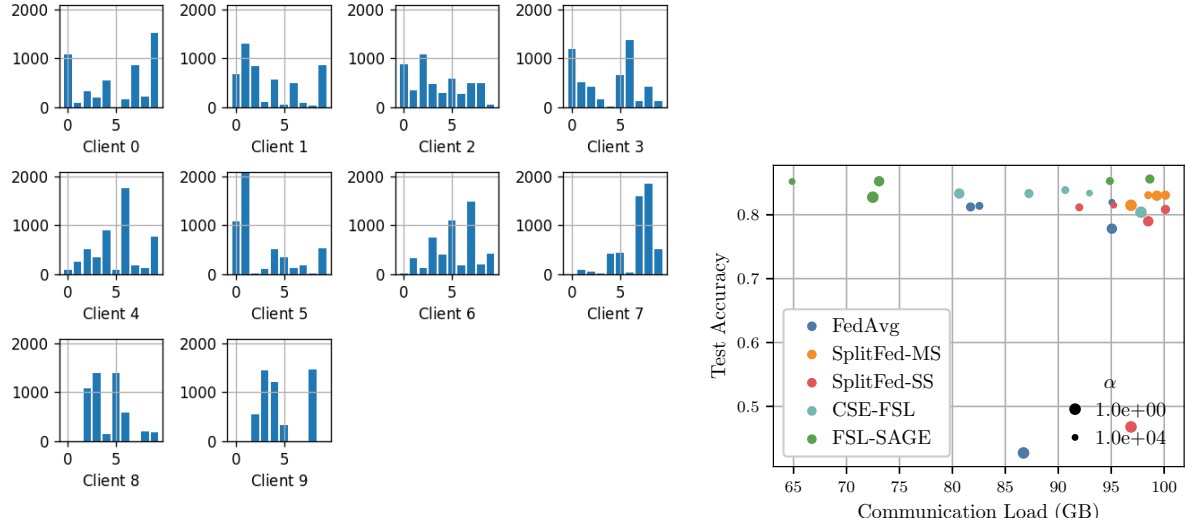

*Figure 8.* Class label distribution for Dirichlet distributed data with $\alpha = 1$ simulating a heterogeneous data distribution across clients.

*Figure 9.* Scatter plot of best accuracy vs. communication load for ResNet-18/CIFAR-10 on 10 clients.

## C. Notes on experimental setup

**Dirichlet Sampling** We use the Dirichlet distribution to simulate client data heterogeneity. Specifically, we sample the proportion of class labels in each client from a Dirichlet distribution, and then sample examples uniformly from each class per client while respecting the proportions sampled from the Dirichlet distribution. The parameter $\alpha$ controls the degree of heterogeneity with smaller values indicating a higher variation in the proportion of class labels distributed across the clients. In Fig. 8, we show an instance of class label distribution for $\alpha = 1$ for CIFAR-10.

**Note on the cut-layer and auxiliary models:** There are two important choices in the design of FSL-SAGE: the choice of the cut-layer and the choice of auxiliary models. *1) Cut-Layer:* The location of the cut-layer determines the communication cost of transmitting the cut-layer features. If the cut-layer features are too large compared to the size of the auxiliary models, the communication gain of FSL-SAGE is not too large compared to CSE-FSL, since both methods expend resources in transmitting the cut-layer features. Although, note that FSL-SAGE does strictly better than CSE-FSL in terms of communication cost. *2) Auxiliary:* For our experiments, we choose auxiliary models as small subsets of the server model for training. This arbitrary choice is able to demonstrate competitive performance for us, but it is important to note that the size of the chosen auxiliary can impact the communication advantage of our method. We perform ablation experiments demonstrating the effect of auxiliary model size on final test performance in Section D.4.

## D. Additional Experiments

### D.1. Accuracy-Communication performance

In Fig. 9 we plot the best accuracy achieved and the corresponding communication cost incurred by the five algorithms for various values of Dirichlet $\alpha$. Each point represents a different (algorithm, Dirichlet-$\alpha$) combination. The best performing algorithms, which maximize accuracy while minimizing communication costs, would be found at the north-west corner of the plot. The radius of the points in the plot, indicates the degree of heterogeneity, with larger points corresponding to lower $\alpha$ and hence more non-i.i.d. data. We observe that FSL-SAGE is able to achieve a high level of accuracy for all tested levels of heterogeneity, being the only method to occupy the north-west corner of the plot, even for certain high degrees of heterogeneity.

### D.2. Latency Analysis

In Fig. 10, we analyze the differences in the computation and communication latencies of the various FL methods used in our CIFAR-10/ResNet-18 experiments. Note that the computation latency of FSL-SAGE is higher than CSE-FSL due to the auxiliary alignment process, which can is computationally expensive for the server. However, the communication

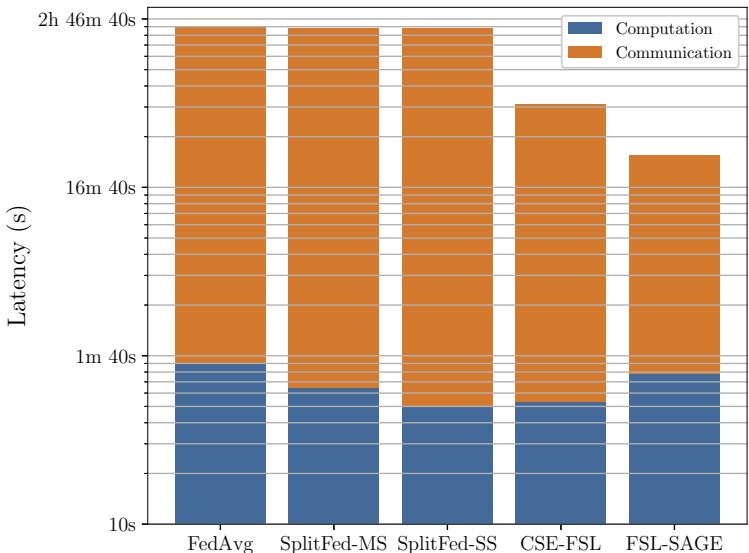

*Figure 10.* Computation and communication latency in various FL methods for 200 rounds. (Note that the y-axis is in logarithmic scale.)

latency of FSL-SAGE is about half that of CSE-FSL, since CSE-FSL requires the server to send the auxiliary models back to the clients after aggregation, while FSL-SAGE only transmits the updated auxiliary models once from the $S$-server to the clients. The communication latency of the three baselines, FedAvg, SplitFed-MS and SplitFed-SS is high because full model transmission for aggregation, and smashed data transmission at every iteration are very communication intensive.

### D.3. Effect of alignment interval $l$ (Ablation study)

In Figures 11 and 12 we plot the final test accuracy of FSL-SAGE using ResNet-18 on CIFAR-10 and CIFAR-100 datasets respectively, for 200 rounds of training. We vary the alignment interval $l$ from once in 2 rounds to once in 10 rounds. Note that, interestingly, there is no clear pattern between $l$ and the final test accuracy for the CIFAR-10 dataset, likely because CIFAR-10 is not a difficult dataset for ResNet-18 to learn. Hence, the server-side model is not very complex, and the auxiliary model learns to mimic it fairly quickly, thus requiring a far fewer update frequency. On the other hand, for a more difficult dataset like CIFAR-100, we see a clear pattern of deterioration of test accuracy with increasing $l$. The alignment interval $l$ is an important trade-off parameter in FSL-SAGE, as it trades-off increased computational load on the server with the final test performance. For our main results, we chose a value of $l = 10$, which is a suitable compromise between the two.

### D.4. Effect of auxiliary model size (Ablation study)

In this section, we study the effect of the size of the auxiliary model on the performance of the trained model. We use the CIFAR datasets, and the ResNet-18 model split into a client-side model and a server-side model as previously described in our main experiments. The client-side model comprises the first two blocks of ResNet-18, and the last two blocks with the final fully-connected layer are used as the server-side model. We try out the following four variations in the auxiliary model:

1. **Linear**: with only the final fully-connected layer of ResNet-18. The size of the auxiliary model is only $\approx 5\,\text{kB}$.

2. **Half**: with $3^{\text{rd}}$ and $4^{\text{th}}$ of ResNet-18, but with only half the original number of layers, i.e. 1 layer per block, and the final fully connected layer. In this case, the size is $\approx 3.5\,\text{MB}$.

3. **Ours**: with only the $3^{\text{rd}}$ block and the fully connected layer. The size for this case is $\approx 8\,\text{MB}$.

4. **Full**: with the auxiliary model being the same architecture as the server-side model. The size is $\approx 40\,\text{MB}$.

In Figures 13 and 14, we see the effect of using the above four auxiliary models on the CIFAR-10 and CIFAR-100 datasets.

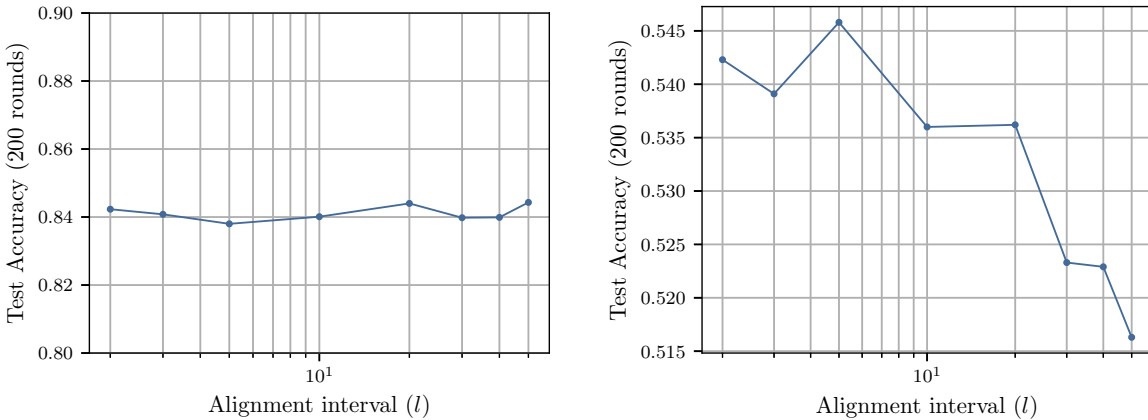

*Figure 11.* Effect of auxiliary model size on final test performance of CIFAR-10 after 200 rounds.

*Figure 12.* Effect of auxiliary model size of final test performance of CIFAR-100 after 200 rounds.

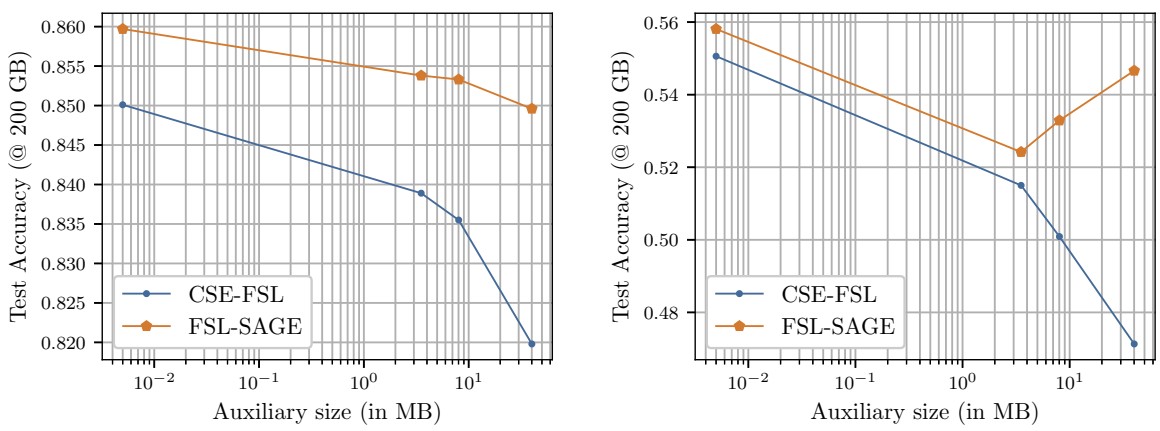

*Figure 13.* Effect of ResNet-18 auxiliary model size on CIFAR-10.

*Figure 14.* Effect of ResNet-18 auxiliary model size on CIFAR-100.

For CIFAR-10, the increasing auxiliary model size shows a drop in test performance. We attribute this to the fact that CIFAR-10 is a relatively simple classification problem, and perhaps a small auxiliary model is able to replicate the nuances in the function learned by the server-side model, while a larger auxiliary model overfits the patterns on the training data. In fact, both CSE-FSL and FSL-SAGE, are able to learn a simpler linear auxiliary model much quicker resulting in a good performance for the linear model in all cases. More importantly, note that the difference in performance of the *same* auxiliary model increases between the two algorithms as the model size increases. This shows that FSL-SAGE is able to learn a more complex auxiliary model more efficiently than CSE-FSL. In Fig. 14, the performance of the FSL-SAGE improves with increasing auxiliary model sizes, in contrast to CSE-FSL.

