# OpenReview forum: "FSL-SAGE: Accelerating Federated Split Learning via Smashed Activation Gradient Estimation"
_ICML.cc/2025/Conference — ICML 2025 poster_

### Official Review · Reviewer_vMHm · 2025-03-10

**Overall Recommendation:** 3

**Summary:**

The paper introduces FSL-SAGE, a framework designed to address the limitations of Federated Learning (FL) and Split Learning (SL). FL struggles with training large models due to client-side memory constraints, while SL incurs high communication latency due to sequential processing. FSL-SAGE combines the data parallelism of FL with the model-splitting efficiency of SL by employing client-side auxiliary models to estimate server-side gradients. These auxiliary models are periodically aligned with the server to minimize communication overhead while preserving accuracy.

Algorithm: Clients use auxiliary models to estimate server gradients locally, enabling parallel training and reducing reliance on frequent server communication. A "lazy" variant further reduces communication by freezing auxiliary models after alignment.
Convergence Guarantees: Theoretically, FSL-SAGE achieves an O(1/√T) convergence rate, matching FedAvg, despite reduced communication costs.
Empirical Results: Experiments on ResNet-18 (CIFAR-10/100) and GPT2-medium (E2E dataset) demonstrate superior accuracy and communication efficiency compared to FedAvg, SplitFed, and CSE-FSL. FSL-SAGE reduces communication costs by up to 10× while maintaining robustness to data heterogeneity.

**Claims And Evidence:**

Yes.

**Essential References Not Discussed:**

Many recent works have explored issues related to client-server communication and convergence in SFL; however, these aspects are not addressed in the manuscript.

[1] Lin, Zheng, et al. "Adaptsfl: Adaptive split federated learning in resource-constrained edge networks." arXiv preprint arXiv:2403.13101 (2024).
[2] Wu, Wen, et al. "Split learning over wireless networks: Parallel design and resource management." IEEE Journal on Selected Areas in Communications 41.4 (2023): 1051-1066.
[3] Oh, Seungeun, et al. "Locfedmix-sl: Localize, federate, and mix for improved scalability, convergence, and latency in split learning." Proceedings of the ACM Web Conference 2022. 2022.
[4] Lin, Zheng, et al. "Hierarchical split federated learning: Convergence analysis and system optimization." arXiv preprint arXiv:2412.07197 (2024).

**Experimental Designs Or Analyses:**

The experimental designs and analyses exhibit both strengths and limitations in validity:
The use of Dirichlet-sampled non-IID data** rigorously tests robustness to heterogeneity, and communication cost metrics align with real-world federated constraints.
Limitations:
1. Experiments on ResNet-18 (edge-deployable without splitting) and GPT-2-medium (outdated for modern LLM scales) fail to validate claims about "large-model" efficacy.
2. The auxiliary model architecture (a subset of the server model) is fixed without ablation studies. This leaves uncertainty about how auxiliary design (e.g., depth, width) affects performance, especially for larger models.
3. The alignment interval l = 10 is arbitrary; no sensitivity analysis explores how varying l impacts accuracy or communication trade-offs.
4. While FedAvg and SplitFed are included, newer hybrid FL-SL methods (e.g., AdaptSFL[1], CPSL[2]) are absent, raising questions about comparative novelty.

[1] Lin, Zheng, et al. "Adaptsfl: Adaptive split federated learning in resource-constrained edge networks." arXiv preprint arXiv:2403.13101 (2024).
[2] Wu, Wen, et al. "Split learning over wireless networks: Parallel design and resource management." IEEE Journal on Selected Areas in Communications 41.4 (2023): 1051-1066.

**Methods And Evaluation Criteria:**

The paper claims that SFL is increasingly relevant for LLM pretraining and fine-tuning in the introduction, yet the experimental validation primarily relies on small-scale models like ResNet-18. This creates a mismatch between the stated motivation and empirical evidence. For instance, ResNet-18 can be easily deployed on edge devices in its entirety, rendering model splitting unnecessary and undermining the practicality of the proposed method for such scenarios. The lack of experiments with prevalent open-source LLMs (e.g., LLaMA 3, or Qwen) weakens the claim about SFL-SAGE’s applicability to modern foundation models. The authors should either substantiate their claims with experiments on larger, cutting-edge LLMs or refine their motivation to align with the evaluated scenarios.

**Other Comments Or Suggestions:**

No.

**Other Strengths And Weaknesses:**

Strengths:
1. The use of an auxiliary model to approximate activation gradients instead of directly computing the loss locally is an innovative approach.
2. The experiments on communication overhead are comprehensive and provide valuable insights.

Weaknesses:
1. The manuscript lacks sufficient experiments on truly large models that necessitate split training. Additionally, the chosen models and datasets are somewhat outdated.
2. The baseline selection is relatively limited, lacking comparisons with closely related studies on SFL.
3. The experiments are insufficient, particularly in terms of exploring the impact of hyperparameters and conducting ablation studies. These aspects should be addressed to strengthen the overall analysis.

**Questions For Authors:**

1. It would be beneficial to indicate the second-best results in the experimental figures, as this could provide additional insights for the result analysis. However, this aspect is not discussed in the manuscript.
2. The manuscript would benefit from a discussion on how system data heterogeneity impacts the final convergence speed and convergence stability. This aspect is crucial for understanding the robustness of the proposed method under varying conditions.

**Relation To Broader Scientific Literature:**

The key contributions of FSL-SAGE are situated within the evolving landscape of federated and split learning, addressing critical gaps identified in prior work. FL, epitomized by FedAvg (McMahan et al., 2016), prioritized data parallelism and privacy but assumed clients could train full models—an impracticality for modern large-scale architectures. Split learning (SL) (Vepakomma et al., 2018) relaxed client memory constraints via model splitting but introduced sequential bottlenecks, as seen in SplitFed (Thapa et al., 2022). Recent efforts like CSE-FSL (Mu & Shen, 2023) reduced communication via local losses but lacked server feedback, risking accuracy degradation. FSL-SAGE bridges these paradigms by introducing auxiliary models to estimate server gradients locally, enabling parallelism while preserving server guidance. The theoretical O(1/√T) convergence rate aligns with FedAvg’s guarantees, demonstrating that split training need not sacrifice convergence speed despite added complexity.

**Theoretical Claims:**

I have carefully examined the correctness of the theoretical proofs and found no apparent errors. The authors have provided sufficient justifications and detailed explanations, ensuring the validity of their claims.

---

> ### Author Rebuttal · Authors · 2025-04-01
>
> We thank the reviewer for their comments.  Please refer to the abbreviations in reviewer **HNw2**'s rebuttal.
>
> > **Comment 1:** Experimental validation relies on small-scale models. Weakens claim that FSL-SAGE applies to large models.
>
> **Response:** We appreciate the reviewer's point about using large-scale LLMs for our experimental evaluation. It is true that smaller models like ResNet-18 or GPT-2m are not large enough to necessitate splitting. However, our goal is to demonstrate the efficacy of our method in relation to other state-of-the-art methods for a given ML model. We make no assumptions on the model architecture or size, so we would expect our theoretical results to hold even for large models. We are very interested to test larger LLMs in our framework, the manual process required for implementing a split architecture and the AM is a challenge for us. However, we didn't have sufficient resources to implement them for larger models in this rebuttal. If this paper is accepted, we will share our source code and welcome collaborations to train larger LLMs.
>
> > **Comment 2:** AM architecture is fixed without ablation study. How does AM size affect performance?
>
> **Response:**  Please see our responses to **Comments 4** and **6** by **iZqi**.  We will add an ablation study of the AM size to the revision.
>
> > **Comment 3:** The alignment interval $l = 10$ chosen without ablation study.  Sensitivity analysis missing.
>
> **Response:** We thank both reviewers **bte9** and **vMHm** for raising this point. We will include an ablation study analyzing the impact of $l$ in the next revision.  However, we would like to note that our **Theorem 4.8** has shown the trade-off effect of the alignment interval $l$. In fact, the bound in Eq. (14) applies to every $l^{th}$ communication round. Thus, if $l$ is very large, the algorithm would take longer to convergence, which makes sense intuitively, since the AMs would get aligned infrequently thus mostly misguiding the client-model.  On the other hand, if $l$ is too small the first few iterations are spent in overtraining the AMs to mimic a near randomly initialized SSM. For our experiments, we manually tuned $l$.
>
> > **Comment 4:** Newer hybrid methods like AdaptSFL, CPSL, not included in baselines. Important references missing.
>
> **Response:** From an optimization perspective, AdaptSFL and CPSL are the same as SplitFed and vanilla SL, respectively.  These hybrid methods either change the structure of model splitting or the sequence of parallel or sequential operations in SL, but these do not impact the algorithmic convergence of the optimization of the model.  Also, the communication efficiency of AdaptSFL and CPSL would be a scaled version of vanilla SL or SplitFed, since these methods do not cut down on the communication cost due to client-server messages at every local iteration, which are both included in our baselines. For these reasons, these two works do not represent new baselines. However, we thank the reviewer for pointing out these and additional references and, we will cite them in the revision.
>
> > **Comment 5:** 1. Experiments use outdated models/datasets. 2. Baselines are limited. 3. Insufficient ablation studies.
>
> **Response:** Please see our responses to **Comment 1, 2** and **3**. We plan to strengthen our experiments by adding ablation studies for $l$ and AM size.
>
> > **Comment 6:** Second-best results in experimental figures.
>
> **Response:** Thank you, we will indicate the second-best results in our figures.
>
> > **Comment 7:** How does system/data heterogeneity affect convergence speed?
>
> **Response:** In our theoretical analysis, we allow for system and data heterogeneity via the quantities $\sigma_l$, i.e., the s.t.d. of the mini-batch of data from its true value, and $\sigma_g$, the s.t.d. of the client's loss function from the loss overall loss function. The expressions in **Theorem 4.3, Eqs. (3)** and **4.8, Eq. (14)** indicate the effect of these quantities on the convergence rate. Larger values of these variance terms decrease the rate of convergence, as is the case with other FL methods.  We appreciate the reviewer for raising this point, and we will add these discussions in our revision.

---

### Official Review · Reviewer_iZqi · 2025-03-13

**Overall Recommendation:** 4

**Summary:**

The paper studies split federated learning with a focus on reducing training latency/communication. It uses an auxiliary model to facilitate the computation of cut-layer gradients. To mitigate potential accuracy drop, the paper aligns the auxiliary model with the server-side model periodically. A solid convergence analysis is provided for the proposed algorithm, and some experiments are given to showcase the effectiveness.

**Claims And Evidence:**

The claims are provided clearly and convincingly with theoretical and/or numerical evidence.

**Essential References Not Discussed:**

Since the main contribution is a convergence analysis for split federated learning, the authors are suggested to include the following references on split (federated) learning convergence, and discuss the differences/improvements.

- Li, Yipeng, and Xinchen Lyu. "Convergence analysis of sequential federated learning on heterogeneous data." NeurIPS, 2023.
- Han, Pengchao, et al. "Convergence analysis of split federated learning on heterogeneous data." NeurIPS, 2024.

**Experimental Designs Or Analyses:**

The experimental results well presented and analyzed. While the reviewer appreciates the theoretical contributions, more experiments (in particular ablation studies) are needed to demonstrate the effectiveness of the approach.

- Han et. al 21 seems to be the most relevant work to this submission. However, it has not been included in the benchmark. If I understand correctly, FSL-SAGE is a generalized version of that in Han et. al 21, where one could use $l=\infty$. At least an ablation study of different choices of $l$ should be included.

- In line 369, the authors ``arbitrarily'' chose the structure of the auxiliary model. This seems to be inappropriate, as one can imagine this choice will have a non-trivial impact on the model performance. Also, the model cut strategy should be investigated empirically as well.

- It is suggested to include training latency (e.g., in wall clock time) as a metric.

- Sec. 4.2-4.3 included more assumptions to establish the convergence results, and those assumptions should be empirically justified. For example, can the authors quantify the value of $\epsilon$ for the proposed algorithm?

**Methods And Evaluation Criteria:**

The choice of models and datasets is suitable and is widely adopted in other related work on (split) federated learning.

**Other Comments Or Suggestions:**

The current version of Fig. 1 is a bit confusing, and the authors may consider adding steps/ordering.

The paper could benefit from proofreading and correcting a list of typos. Here are a few examples.

- In line 71, ``due'' -> ``due to''
- In line 396, CIFAR-100 should be corrected to be CIFAR-10.

It would be better to include codes for reproducibility purposes.

**Other Strengths And Weaknesses:**

- The paper implemented LoRA fine-tuning of LLMs in a split federated learning environment, which is very relevant and could benefit future research efforts.

**Questions For Authors:**

Can the authors specify what technical challenges model alignment brings to the proofs and/or implementations compared to those in Han et. al 21?

**Relation To Broader Scientific Literature:**

- A convergence analysis is provided for split federated learning with client-side update using auxiliary models.

**Theoretical Claims:**

I have quickly checked the proofs of the theorems, and no obvious errors have been spotted.

However, there are some issues.
- In Theorem 4.3 Eq. (3), it seems that $c$ is a typo (originally defined for clients). Can the authors specify its physical meaning?
- Theorem 4.3 is agnostic to $l$, i.e., how frequently the auxiliary models are aligned with the server-side model. Does this mean the convergence is not affected by $l$, or the bound is loose?

---

> ### Author Rebuttal · Authors · 2025-04-01
>
> We thank the reviewer for their comments.  Please refer to the abbreviations in reviewer **HNw2**'s rebuttal.
>
> > **Comment 1:** In Theorem 4.3, physical meaning of $c$?
>
> **Response:** Thank you, we missed defining the constant $c$ in **Theorem 4.3**, but it is defined later in **Theorem 4.8**. We will add the definition in **Theorem 4.3** in the next revision. $c < 0.5 - 20K^2 \eta_L^2 \widehat{L}_c^2$ is a positive constant defined to simplify the final expression of convergence.
>
> > **Comment 2:** Theorem 4.3 is agnostic to $l$; does convergence depend on $l$?
>
> **Response:** **Theorem 4.3** is indeed agnostic to $l$, the alignment interval, not because the convergence is unaffected by $l$ or that the bound is loose, but because $l$ directly affects the _auxiliary estimation error_, $\varepsilon^t$, given in **Theorem 4.3**, left-hand-side of Eq. (5). Our final result in **Theorem 4.8** Eq. (14) is obtained by further bounding $\varepsilon^t$. There, we have a bound on every $l^{th}$ round, which implies that a larger $l$ would need a proportionally larger number of rounds $T$ (i.e., slower) to converge and vice-versa.
>
> > **Comment 3:** [1] though most relevant, is not included in baselines.
>
> **Response:** We did not include [1] in our baselines for the following reasons:
> * [1] uses multiple copies of the SSM, one for each client, and also uploads cut-layer activations to the server at every iteration, making it very memory and communication inefficient.
> * In CSE-FSL [2], which is a baseline for our work, the authors demonstrated that they performed much better than Han et. al [1] in terms of communication efficiency. Thus, it suffices to compare with [2].
> * Lastly, our method is not a generalization of [1]. The following are two important differences between our method and [1]:
>     1. [1] updates the AMs via local loss functions at each local iteration, while we do not.  This means that even as $l\to\infty$, our method will behave very differently.
>     2. Han et. al [1] used multiple SSMs, one for each client, while we use only 1 SSM.
>
> [1] Han, Dong-Jun et al. “Accelerating Federated Learning with Split Learning on Locally Generated Losses.” (2021).
> [2] Mu, Yujia, and Cong Shen. "Communication and storage efficient federated split learning." ICC 2023-IEEE International Conference on Communications. IEEE, 2023.
>
> > **Comment 4:** Arbitrary choice of AMs has non-trivial impact on performance. Cut strategy should also be investigated.
>
> **Response:** The choice of an appropriate AM architecture and cut-layer are interesting problems in their own right, but somewhat out of scope of this work. Like all other SL methods [1-2], we focus on a given split and our work works with **general models**. Similarly, we use a given AM and compare to CSE-FSL [2]. As we mention in **Comment 6** below, we will include an ablation study on AM sizes in the next revision.
>
> > **Comment 5:** Training latency as a metric.
>
> **Response:** We will estimate the latency assuming a fixed communication and computation bandwidth between the clients and server. We will include this simulated wall-clock result to the revised manuscript.
>
> > **Comment 6:** Empirically justify $\epsilon$ in learnability assumption.
>
> **Response:** This is also a response to **bte9**'s **Comment 3**.  It is intractable to check if a given AM is in-expectation PAC learnable or not.  However, in our experiments, an AM which is chosen by taking a small portion (of size $\leq 0.1\times$) of the respective SSM demonstrates comparable, if not better convergence to the tested baselines. We would also like to note that **Assumptions 4.6** and **4.7** are sufficient but not necessary conditions for convergence.  In our revision, we will add an ablation study of the AM size on performance. There, we will also test smaller AMs than the ones presented in the paper.
>
> > **Comment 7:** Include references for convergence of SplitFed.
>
> **Response:** We thank the reviewer for pointing out the references on convergence analyses of sequential FL and SplitFed. We will add these to the next revision.
>
> > **Comment 8:** Some typos; inclusion of source code.
>
> **Response:** We thank the reviewer for detecting these oversights.  In the next revision we will:
> * Proofread and fix typos.
> * Add numbers to **Fig. 1** to indicate the order of operations.
>
> We will also share our source code after the review process (we have already shared our code as supplementary material in a `.zip` file for the review process).
>
> > **Comment 9:** Technical challenges faced in proofs and implementation.
>
> **Response:** While [1] originally introduced the idea of using auxiliary models to train clients in parallel, our work involves aligning the auxiliary model directly to _mimic_ the server-side model. This brings about several technical challenges in convergence analysis, all of which are detailed under the heading **2) Technical Challenges, Lines 70-90** in our manuscript.

---

> > ### Comment · Reviewer_iZqi · 2025-04-02
> >
> > My comments have been satisfactorily addressed. I am ready to raise my score to 4.

---

> > > ### Author Response · Authors · 2025-04-07
> > >
> > > We thank the reviewer for their valuable feedback, and for helping us improve the quality of our manuscript.

---

### Official Review · Reviewer_HNw2 · 2025-03-14

**Overall Recommendation:** 3

**Summary:**

The paper addresses the computational burden faced by clients when performing local updates on whole (possibly large) models by splitting the model into server-side and client-side components, following the approach used in prior split learning (SL)-based methods. Unlike previous works, the authors propose an approach to estimate gradient feedback from the server using an auxiliary model on the client side, which significantly reduces latency and communication overhead.
Specifically, the true cut-layer gradient is periodically received from the server, and the auxiliary model is trained using an MSE loss to approximate the gradient. The paper provides theoretical convergence analysis and demonstrates the broad applicability of their method by conducting the experiment on language model as well.

**Claims And Evidence:**

The claims in this study are supported by experimental evaluations and detailed analyses.

**Essential References Not Discussed:**

I have not noticed any key references that were missing from the paper.

**Experimental Designs Or Analyses:**

The experiments and evaluations are well designed and valid.

**Methods And Evaluation Criteria:**

The proposed method is conceptually well-grounded, and the evaluation is appropriate.

**Other Comments Or Suggestions:**

I appreciate the careful consideration of the alignment mechanism in the convergence analysis, and the attempt to analyze the lazy version as well. These aspects add significant value to the paper.

Regarding the convergence rate, while the rate with respect to $T$ matches existing results, one of the key focuses is how the rate is expressed in terms of the number of clients. In fact, a linear speedup with respect to the number of clients is often achieved in recent FL works. However, in the convergence presented in this paper, it is unclear how this aspect is handled.

Additionally, while uploading the alignment dataset itself poses a potential privacy leakage risk, this issue is not discussed in the paper. Including a discussion on this aspect would strengthen the paper.

**Other Strengths And Weaknesses:**

### Strengths

- The proposed method effectively addresses the challenges of existing SL-based approaches by introducing an auxiliary model to approximate gradient feedback.
- The gradient feedback estimation mechanism significantly reduces communication overhead and latency while ensuring high performance.
- A convergence analysis is provided, covering both FSL-SAGE and its lazy version.
- Experimental results on language model are presented, demonstrating the broad applicability of their method.

### Weaknesses
- Sending the alignment dataset to the server poses a potential risk of privacy leakage.
- The auxiliary model is not particularly small. it appears to be even larger than the client-side split model, which may impose additional computational burden on the client compared to traditional SL-based methods.
- From the server-side perspective, the optimization of $x_a$ is required for 'all' the clients. As suggested by the equation (2), if I understand correctly, this optimization might require double derivative with respect to $x_a$, which could introduce a significant computational burden on the server.

**Questions For Authors:**

Please refer to the weaknesses and comments parts.

**Relation To Broader Scientific Literature:**

The paper builds on prior SL based methods for addressing key challenges in FL, e.g., computational efficiency, communication overhead, and latency. The paper introduces an auxiliary model for gradient feedback estimation, significantly reducing latency and communication costs. The authors also provide rigorous convergence analysis for both FSL-SAGE and its lazy variant, ensuring theoretical foundation of the paper.

**Theoretical Claims:**

The convergence bound is rigorously analyzed, including both FSL-SAGE and its lazy variant.

---

> ### Author Rebuttal · Authors · 2025-04-01
>
> We thank the reviewer for their comments.  For brevity we will use the abbreviations FSL for federated split learning, FL (SL) for Federated (resp. Split) Learning, CSM (SSM) for client-side (resp. server-side) model, and AM for auxiliary model.
>
> > **Comment 1:** Sending alignment dataset to server poses potential privacy risk.
>
> **Response:** We agree that there are privacy concerns about uploading cut-layer data to the server, but one can argue that this risk would be equally present in any SL algorithm which sends smashed data to the server. The study of attacks and privacy guarantees for our algorithm is indeed important, which deserves a separate paper and is beyond the scope of this manuscript. Also, we note that the FSL setting in this paper is the same as in other SL algorithms published in the literature, hence having the same privacy performance.
>
> > **Comment 2:** The AM size is larger than the CSM; imposes compute burden on clients.
>
> **Response:**  We agree that the AMs used in our experiments are large.  Please see our response to **Comment 6** of reviewer **iZqi**. We will conduct a more thorough ablation study to demonstrate the effect of AMs size in our revision.
>
> > **Comment 3:** Optimization of $x_a$ requires double derivative, causing compute burden on server.
>
> **Response:** It is true that the alignment of AMs requires a double derivative, and this could pose a computation burden, especially if we use large AMs. Our ablation experiments on the AMs, as discussed above, will help clarify if this is indeed a significant burden.
> In our experiments with GPT2m, we used an AM of size 92.4M and didn't face very high compute burdens.
>
> > **Comment 4:** In theoretical results, does FSL-SAGE enjoy linear speedup?
>
> **Response:** Thank you for raising this interesting point, it will help clarify our manuscript. Unfortunately all FSL methods using a single server-model, including our method, which handles one client at a time, lacks linear speedup [1]. Primarily, this is because the server-side model is trained sequentially on cut-layer activations received from the clients. We will add a note about this aspect to the revision.
>
> [1] Han, Pengchao, et al. "Convergence analysis of split federated learning on heterogeneous data." arXiv preprint arXiv:2402.15166 (2024).

---

### Official Review · Reviewer_bte9 · 2025-03-14

**Overall Recommendation:** 1

**Summary:**

The paper proposed a new federated split learning algorithm called FSL-SAGE. It builds upon existing works on local split federated learning, where client updates are derived from local approximations using an auxiliary model attached to each client. A key challenge with previous approaches is that, due to the lack of feedback from the server, these local auxiliary modules can drift away from the optimal solution. FSL-SAGE addresses this issue by periodically aligning these auxiliary modules via an additional optimization loop conducted on the server side.

**Claims And Evidence:**

The core contribution of this work is rather marginal. The core element of this technique is the server feedback to the client’s auxiliary network which is carried out indirectly via periodic alignment of these modules on the server-side is straightforward and has been discussed before.

**Essential References Not Discussed:**

Since FSL-SAGE involves sending smashed data (activations) and labels to the server, the paper should reference works that analyze privacy risks and provide mitigation approaches.

The periodic alignment of these auxiliary modules in the centralized setting is discussed in a related paper LSL-PGG[1] which should be cited.

[1] Bhatti, Hasnain Irshad, and Jaekyun Moon. "Locally Supervised Learning with Periodic Global Guidance." arXiv preprint arXiv:2208.00821 (2022).

**Experimental Designs Or Analyses:**

1. The paper does not analyze the computational load on the server side, which is crucial for large-scale federated learning scenarios where thousands of devices participate simultaneously. Storing alignment datasets for every client and running the secondary optimization loop might create a bottleneck on the server.

2. Also, the added computation due to increased auxiliary network size as compared to small aux networks in existing techniques should be discussed.

3. The paper does not provide guidance for practitioners on how to select the optimal hyperparameters introduced e.g. the alignment period, neither does the paper provide an ablation study.

4. Given that the key hypothesis is that periodic alignment is what makes the difference, an analysis of auxiliary gradient error over time is essential to confirm it.

5. Wall-clock time comparison or an estimate would be great to understand the potential latency introduced by the alignment overhead.

**Methods And Evaluation Criteria:**

The authors use decent datasets and large models for evaluation, and the baselines are generally appropriate. However, the setting of a single local epoch is limiting and not widely practiced. Additionally, the manual nature of model splitting may hinder generalization to other real-world scenarios. Moreover, the number of clients used in the experiments is very low, whereas typical federated learning setups involve hundreds of clients with partial participation.

**Other Comments Or Suggestions:**

The paper has some typos, e.g. “limtations” and “sever-side”. The text “For LoRA finetuning” can be misleadin, consider clarifying the context.. A brief section mentioning limitations would be helpful. Also, readers might wonder: does sending labels to the server compromise privacy? A line addressing this could be added for reader to understand the potential risk associated with FSL techniques.

**Other Strengths And Weaknesses:**

1.	A potential bottleneck for the system is when client partially participates. In the low participation ratio, the server still has to store alignment datasets for every client. Also it is unclear how outdated data is managed when clients drop out.
2.	The choice of the split cut-layer determines the size of the smashed activation, which can be substantial. In this method, the server must store these activations for each client for alignment, potentially imposing a significant computational and memory burden. A more detailed analysis of this aspect is needed.
3.	The size of the auxiliary network is quite large compared to existing baselines. For ResNet-18, the auxiliary network (2.1M parameters) is nearly three times the size of the client’s original network, and it is even larger (92.4M parameters) for the language task. I would like to see how does it compare to the baselines when the aux networks are used as suggested by those works.

**Questions For Authors:**

1.	Does the client-side auxiliary module also get updated during the local update step?
2.	What happens if the auxiliary network is smaller? Is there a minimum size required for the auxiliary network to effectively serve as a surrogate for the server model?
3.	Can you provide more insight or data on choosing the alignment interval? The theory gives a trade-off, but practically how does it affect the performance.
4.	How does the server scale when the number of clients increases significantly (e.g., hundreds or thousands)?
5.	How do you justify the trade-off between increased auxiliary network size and overall performance, especially considering the potential burden on client devices?

**Relation To Broader Scientific Literature:**

The paper is well-positioned within the federated and split learning literature. However, it would benefit from discussing privacy analyses and through evaluation of the trade-offs introduced.

**Theoretical Claims:**

1. The proof sketch appears technically sound, and most assumptions are standard. However, the in-expectation PAC learnability assumption, though theoretically mild, may be challenging in practice since it requires the auxiliary model to be sufficiently expressive (large enough).

2. In previous works, the auxiliary network was kept small because it was not intended to fully replace the server model, whereas in this paper, a larger auxiliary network is used, even larger than the client model itself.

3. Additionally, the paper assumes an honest server, yet in practice, the server could be prone to various attacks. A discussion on potential attacks, particularly regarding privacy concerns, is missing.

---

> ### Author Rebuttal · Authors · 2025-04-01
>
> Thank you for your comments and suggestions. Due to space limitation, we could only respond to a subset of more critical comments in this rebuttal.  But we are happy to continue to complete our responses to your remaining comments in the discussion stage when new space opens up. Please refer to the abbreviations in **HNw2**'s rebuttal.
>
> > **Comment 1:** Marginal contribution; server feedback already mentioned in [1]. Periodic alignment is related to LSL-PGG[1] which should be cited.
>
> Thank you for pointing us to Ref. [1].  We would like to clarify that our core contributions are distinct from Refs. [1-2] as follows:
>
> * Unlike LSL-PGG [1], which is a centralized algorithm with a goal to reduce training memory footprint, our work, is a **federated** split learning algorithm designed to train large models using FL on data distributed over commodity hardware.
>
> * In [1] the AMs are updated indirectly via a global loss update.  In contrast, we update the AMs directly to _mimic_ the SSM by minimizing the MSE loss.
>
> * To the best of our knowledge, our approach is the first AM based FSL approach to have a convergence guarantee on the joint model. Previous works [2] only provide separate convergence guarantees for the CSM and SSM, which do not imply convergence of the join model to a minimum.
>
> We will cite [1] and clarify these differences in the next revision
>
> [1] Bhatti el al. "Locally Supervised Learning with Periodic Global Guidance." arXiv preprint arXiv:2208.00821 (2022).
> [2] Mu el al. "Communication and storage efficient federated split learning." ICC 2023-IEEE International Conference on Communications. IEEE, 2023.
>
> > **Comment 2:** Single local epoch setting, small number of clients $m$, and manual splitting impractical.
>
> 1. **Single Epoch:** We chose the local epoch to be 1 in our experiments primarily for a fair comparison to previous literature [2], which also has the local epoch as 1. To clarify, our algorithm statement and theoretial analysis makes **no such assumption** regarding the number of local iterations or local epochs.
>
> 2. **Manual Splitting:** Our framework does **not** depend on the splitting technique used, and we do **not** used the term 'manual splitting' anywhere in our manuscript. Developing an algorithm that can optimally split a model would indeed be very interesting, but this fundamental topic deserves a seperate paper that is beyond the scope of this manuscript. We will pursue this topic in our future studies and we thank the reviewer for pointing out this direction.
>
> 3. **Number of Clients:** Our theoretical analysis reveals that $m$ in the FL setup does not impact our convergence rate. We have only simulated 10 clients in our experiments due to resource limitations.  We note that our shared source code is configurable to accomodate for much higher $m$ given enough GPU resources.
>
> > **Comment 3:** 1. In-Expectation learnability assumption challenging in practice. 2. AMs are larger that the CSMs adds to client computation. What is the trade-off between AM size and performance? 3. How do we compare to baselines when smaller AMs are used.
>
> * Please see our response to **Comment 6** by **iZqi**.
> * In the next revision, we will also compare our method against baselines using smaller AMs as you suggested.
>
> > **Comment 4:** Assumes honest server; Include discussion of potential privacy concerns. Cite works analyzing privacy risks. Does sending labels to server compromize privacy?
>
> * Please see our response to **Comment 1** by **HNw2**. Thank you for pointing us to these references. We will cite them in our revision as you suggested.
> * Sharing labels is the same in all FSL methods. Hence, our method achieves the same privacy performance as in other FSL methods.
>
> > **Comment 5:** 1. Server computational load in large-scale FL; Storage and computation bottleneck due to alignment. 2. How does server deal with partial client participation and client drop-out?
>
> Storing the alignment dataset at the server and alignment can indeed be a bottleneck at the server. To mitigate the need for storage, a simple solution is to align the AMs on the most recent batch of cut-layer activations on an on-demand basis, thus also solving the problems associated with client drop-out and partial client participation. Also, one can perform alignment in another process on the server so it doesn't bottleneck the SSM optimization. While these are interesting engineering extensions to our algorithm, they do not affect our theoretical claims and results, which are our main contributions.
>
> > **Comment 6:** Key hypothesis is periodic alignment; analysis of auxiliary gradient error is essential.
>
> Thanks for the comment.  We analyze the auxiliary gradient error given by Eq. (5) in the manuscript, in **Section 4.2**. We can bound this term as $\mathcal{O}(1/\sqrt{T})$ provided the in-expectation learnability assumption holds, which shows that the gradient error decreases with the same sub-linear rate.

---

### Decision · Program_Chairs · 2025-05-01

**Decision:**

Accept (poster)

**Comment:**

Reviewer bte9 gave a strong reject score and it seems that the rebuttal did not change his mind. However, other reviewers tend to accept this paper. So, I think I need help fron the SAC.